# Gibbs Sampling with People

**Peter M. C. Harrison**[*]
Max Planck Institute for Empirical Aesthetics
Frankfurt
peter.harrison@ae.mpg.de

**Raja Marjieh**[*]
Max Planck Institute for Empirical Aesthetics
Frankfurt
raja.marjieh@ae.mpg.de

**Federico Adolfi**
Max Planck Institute for Empirical Aesthetics
Frankfurt
federico.adolfi@ae.mpg.de

**Pol van Rijn**
Max Planck Institute for Empirical Aesthetics
Frankfurt
pol.van-rijn@ae.mpg.de

**Manuel Anglada-Tort**
Max Planck Institute for Empirical Aesthetics
Frankfurt
manuel.anglada-tort@ae.mpg.de

**Ofer Tchernichovski**
Hunter College CUNY
The CUNY Graduate Center
otcherni@hunter.cuny.edu

**Pauline Larrouy-Maestri**
Max Planck Institute for Empirical Aesthetics
Frankfurt
pauline.larrouy-maestri@ae.mpg.de

**Nori Jacoby**
Max Planck Institute for Empirical Aesthetics
Frankfurt
nori.jacoby@ae.mpg.de

[*]Equal contribution.

## Abstract

A core problem in cognitive science and machine learning is to understand how humans derive semantic representations from perceptual objects, such as color from an apple, pleasantness from a musical chord, or seriousness from a face. Markov Chain Monte Carlo with People (MCMCP) is a prominent method for studying such representations, in which participants are presented with binary choice trials constructed such that the decisions follow a Markov Chain Monte Carlo acceptance rule. However, while MCMCP has strong asymptotic properties, its binary choice paradigm generates relatively little information per trial, and its local proposal function makes it slow to explore the parameter space and find the modes of the distribution. Here we therefore generalize MCMCP to a continuous-sampling paradigm, where in each iteration the participant uses a slider to continuously manipulate a single stimulus dimension to optimize a given criterion such as 'pleasantness'. We formulate both methods from a utility-theory perspective, and show that the new method can be interpreted as 'Gibbs Sampling with People' (GSP). Further, we introduce an aggregation parameter to the transition step, and show that this parameter can be manipulated to flexibly shift between Gibbs sampling and deterministic optimization. In an initial study, we show GSP clearly outperforming MCMCP; we then show that GSP provides novel and interpretable results in three other domains, namely musical chords, vocal emotions, and faces. We validate these results through large-scale perceptual rating experiments. The final experiments use GSP to navigate the latent space of a state-of-the-art image synthesis network (StyleGAN), a promising approach for applying GSP to high-dimensional perceptual spaces. We conclude by discussing future cognitive applications and ethical implications.

# 1  Introduction

Humans continuously extract semantic representations from complex perceptual inputs, re-expressing them as meaningful information that can be efficiently communicated to primary cognitive processes such as memory, decision-making, and language [1–3]. Effective semantic representation seems to be a prerequisite for intelligent behavior, and is correspondingly a core topic of study in both cognitive science and machine learning [4–6].

One way of studying semantic representations in humans is to present participants with stimuli that exhaustively sample from a stimulus space (e.g., the space of visible colors) and ask them to evaluate these stimuli (e.g., [7]). Unfortunately, this method works poorly for high-dimensional stimuli whose parameter spaces are too large to explore exhaustively. An alternative approach is to hand-construct stimulus sets to test specific hypotheses about semantic representations (e.g., that slow melodies tend to sound sad, [8]); however, this approach relies heavily on prior domain knowledge, and is poorly suited to exploratory research.

Markov Chain Monte Carlo with People (MCMCP) addresses this problem [9–11]. MCMCP takes as input a stimulus space (e.g., visible colors) and a target semantic category (e.g., 'danger'). In each trial, participants are presented with two stimuli and are asked which comes from the category. By virtue of MCMCP's adaptive procedure, stimulus selection becomes progressively biased towards parts of the stimulus space that represent the category. The resulting process iteratively characterizes the subjective mapping between the stimulus space and the semantic concept for a given participant or participant group. The technique provides a way for cognitive scientists to systematically quantify subjective aspects of perception, for example the way in which participants from a particular musical culture hear certain chords as 'consonant', or the way in which participants have certain subjective ideas of what a 'serious' face looks like. The approach has been shown to outperform reverse correlation, a related non-adaptive method for mapping semantic categories to perceptual spaces [12].

According to the underlying theory, MCMCP converges asymptotically to the participant's internal probabilistic representation of a given semantic category within a stimulus space [9]. However, its practical convergence rate is limited for several reasons. The first concerns the response interface: MCMCP is traditionally limited to binary choice responses, which can only provide a small amount of information per trial (1 bit), much less than the theoretical limit of other response interfaces (e.g., sliders). The second depends on the proposal function that generates successive stimuli: a too-narrow proposal function makes the process slow to find the modes of the distribution, whereas a too-wide proposal function makes it harder for the process to estimate the mode with much precision [13, 14].

Here we present a new technique for addressing these problems, termed Gibbs Sampling with People (GSP). While MCMCP corresponds to a human instantiation of the Metropolis-Hastings MCMC sampler, GSP corresponds to a human instantiation of a Gibbs sampler. Crucially, unlike [15], GSP has participants respond with a continuous slider rather than a binary choice. This has two effects: first, it substantially increases the upper bound of information per trial, and second, it eliminates the need to calibrate a proposal function. We further show how GSP can be formulated in utility theory, thereby generalizing the approach from discrete to continuous semantic representations, and we show how GSP can be shifted towards deterministic optimization through an aggregation process.

This paper continues with a review of MCMCP and a theoretical account of GSP. We then describe four studies applying GSP to various visual and auditory domains, ranging from simple low-dimensional problems to complex high-dimensional problems parameterized by deep neural networks. These studies include experiments directly implementing GSP and MCMCP, control experiments investigating different hyperparameters, and validation experiments for the generated outputs. All combined, these 25 experiments represent data from 5,178 human participants.[1]

# 2  Theory

## 2.1  MCMC and MCMCP

MCMC is a procedure for sampling from distributions whose normalization constants are impractical to compute directly. It works by constructing a Markov chain whose stationary distribution corre-

sponds to the probability distribution of interest; given enough samples from this Markov chain, it is then possible to approximate the probability distribution arbitrarily closely.

The MCMC algorithm may be performed by choosing an arbitrary initial Markov state $x$ from the parameter space, then repeating the following steps until convergence: (1) Sample a candidate $x^*$ for the next state of the Markov chain according to some *proposal function* $q(x^*; x)$; (2) Decide whether to accept this candidate according to an appropriate *acceptance function* $A(x^*; x)$ constructed to reflect the probability distribution of interest. In the case of a symmetric proposal function, the acceptance function takes a simple form known as the Barker acceptance function and is given by $A(x^*; x) = \pi(x^*)/(\pi(x) + \pi(x^*))$ where $\pi(x)$ is the target distribution [16].

In MCMCP the acceptance function is replaced with a human participant, whose task is to choose between the current state and the candidate state [9]. The trick is then to frame this task such that the participant's choices correspond to the acceptance function for an interesting probability distribution. The solution presented in the original MCMCP paper is to tell the participant that one of the stimuli comes from a class distribution (e.g., cats), and one of them comes from an unknown category. The authors suppose that the participant computes the posterior probability of class membership assuming a locally uniform likelihood for the alternative class, and then selects a stimulus with probability proportional to its posterior probability of class membership. Under these assumptions, the participant's behaviour can be shown to correspond to the classic Barker acceptance function where the underlying probability distribution is the likelihood function for the class being judged.

This formulation is elegant but it has two important limitations. First, it can only be applied to semantic representations that take categorical forms; the derivation does not make sense for continuous semantic representations (e.g., pleasantness). Second, it assumes that participants make their choices with probabilities equal to their posterior probabilities of class membership (a process termed *probability matching*) as opposed to the Bayes-optimal strategy of deterministically maximizing this posterior probability [17]. Humans do indeed seem to exhibit probability matching in certain contexts, but a convincing cognitive model ought to explain how this sub-optimal process arises [18].

Here we reformulate MCMCP (and later GSP) without these limitations. We suppose that the participant is asked to select the stimulus that best matches a given criterion $C$; for example 'select the most pleasant chord' or 'select the color that most resembles lavender'. We suppose that the participant performs this task by extracting a *utility* value for each stimulus, and selecting the stimulus with the maximum utility [19]. In the case of the class membership tasks typically used in MCMCP, we might hypothesize that the utility corresponds to the subjective likelihood of the stimulus conditioned on the class of interest; however, in the general case the utility function need not necessarily correspond to a probability distribution. The utility value is however assumed to have a deterministic component and a noise component, namely $U_i = \ell_i + n_i$, where $\ell_i$ is the deterministic utility of stimulus $i$, and $n_i$ is the associated noise. This noise component can capture participant-level noise from sensory [20, 21] and cognitive [21, 22] processes, as well as population-level noise corresponding to individual differences in the utility function [19]. In the case where the noise components are i.i.d. and have an extreme value distribution common to discrete choice models, it can be shown that the probability of selecting a given stimulus $s_1$ is equal to $(1 + \exp(-\gamma(\ell_1 - \ell_2)))^{-1}$ where $\gamma^{-1}$ corresponds to the scale parameter of the noise component. If the utility is assigned based on subjective likelihood $\ell_i = \log p(s_i|C)$, then this equation reproduces the Barker acceptance function with target distribution $\pi(s) \propto p^\gamma(s|C)$. This justifies MCMCP for an optimal observer with a noisy utility function (for proof and discussion see Appendix A.1).

## 2.2 Gibbs Sampling with People

Gibbs sampling is an alternative approach for sampling from probability distributions [23], defined as follows. Let $p(z_1, \ldots, z_n)$ be a target distribution over an $n$-dimensional state space from which one would like to sample, and choose a starting vector state $\mathbf{z}^{(1)} = (z_1^{(1)}, \ldots, z_n^{(1)})$. Then, circularly update coordinates by sampling from $p(z_k^{(i+1)}|z_1^{(i+1)}, \ldots, z_{k-1}^{(i+1)}, z_{k+1}^{(i)}, \ldots, z_n^{(i)})$.

In GSP the participant provides the coordinate updates. This is achieved by presenting the participant with a slider that is associated with the current stimulus dimension $z_k$ and instructing the participant to move the slider to maximize a certain criterion, such as the pleasantness of a sound or the resemblance of a fruit to a strawberry. To analyze the decision step, let us discretize the slider into a set of points $\{z_k^i\}_i$ and let $z_{-k}$ denote the other fixed dimensions. As before, suppose that each point on the slider

is associated with a utility that contains both a deterministic $\ell(z_k^i, z_{-k})$ and a noise $n_i$ component, and suppose that the participant chooses the slider location that maximizes the utility. Then, under similar assumptions to those made in the MCMCP case, the probability distribution over slider locations is

$$p(\text{choose } i) = p(z_k^i | z_{-k}) = \frac{e^{\gamma \ell(z_k^i, z_{-k})}}{\sum_j e^{\gamma \ell(z_k^j, z_{-k})}} \quad (1)$$

and as the granularity of the slider tends to infinity, the denominator becomes a marginal, and GSP becomes a sampler from $p(\mathbf{z}) \propto e^{\gamma \ell(\mathbf{z})}$ (for proof and discussion see Appendix A.2).

As with MCMCP, GSP can be used to explore either categorical or continuous semantic representations. In the former case, the experimenter might ask a question like 'adjust the slider until the image most resembles a dog', and the participant's utility function might correspond to the log probability of the image given the class, $\ell(\mathbf{z}) = \log p(\mathbf{z}|C)$; in this case the sampler's stationary distribution will be proportional to $p^\gamma(\mathbf{z}|C)$. In continuous semantic representations, the utility function may not correspond to a probability distribution, and the interpretation would simply be that the sampler explores different regions of the space in proportion to their exponentiated utility $e^{\gamma \ell(\mathbf{z})}$.

The noise parameter $\gamma^{-1}$ is important for the behavior of the sampler. As $\gamma^{-1} \to 0$, the choice distribution becomes increasingly peaked around the highest utility item on the slider, shifting thus the sampler into an optimization regime (specifically, coordinate descent). Typically we are interested in minimizing $\gamma^{-1}$, so as to maximize the utility of the samples (mode seeking); however, some noise is still desirable because it helps drive exploration of the utility space.

There are two main ways to reduce the effective noise, $\gamma^{-1}$. One approach is to estimate the joint distribution $p(\mathbf{z}) \propto e^{\gamma \ell(\mathbf{z})}$ by fitting a kernel density estimate (KDE) to the GSP samples, then simulating $\gamma^{-1} \to 0$ by taking the distribution's mode. For simple distributions, this mode can also be estimated by averaging over samples. However, neither KDEs nor averaging are well-suited to complex high-dimensional spaces, where the joint distribution is hard to estimate reliably.

An alternative approach is to manipulate the Gibbs sampler itself. Specifically, suppose that we collect multiple samples from the conditional distribution of a given step of the Gibbs sampler $p(\text{choose } i) \propto e^{\gamma \ell(z_k^i, z_{-k})}$, and then simulate $\gamma^{-1} \to 0$ by returning the peak of the one-dimensional KDE from these samples (or potentially the sample mean). This will in turn simulate $\gamma^{-1} \to 0$ for the joint distribution $p(\mathbf{z}) \propto e^{\gamma \ell(\mathbf{z})}$ as produced by the Gibbs sampler. The practical advantages of this approach are that (a) we restrict density estimation to a more tractable one-dimensional case, and (b) the same stimulus can be re-used for multiple trials, which can be useful when stimuli are slow to create. We explore both KDE and mean aggregation in this paper.

There are several ways to assign the iterations of a GSP or MCMCP chain to human participants. In *within-participant* chains, the entire chain is completed by just one participant, and the resulting samples reflect the semantic representations of that single participant. In *across-participant* chains, each iteration comes from a different participant, and the samples then reflect shared semantic representations across participants (Fig. S1). While within-participant chains can theoretically be used to study individual differences, here we focus on studying representations at the level of the participant group, using both chain types and averaging over participants where appropriate.

Researchers interpreting MCMCP and GSP results must think carefully both about the definition of the stimulus space and of the participant group. For example, if the stimulus space only includes male voices, the results may not be generalizable to female voices. Similarly, if the participant group comprises solely US participants, then the results may not be generalizable to Japanese participants. Of course, these issues are by no means limited to MCMCP and GSP, but apply rather to the majority of psychological research. We will revisit these matters below.

A related paradigm with a Gibbs sampler interpretation is *serial reproduction*, where one participant's imitation of a stimulus becomes the next stimulus in a transmission chain [24–30]. However, serial reproduction is limited to percepts that can be entirely reproduced in a single trial (e.g., spoken sentences, [30]). In contrast, GSP participants only ever have to manipulate one stimulus dimension at a time, even if the stimulus itself is high-dimensional. This allows GSP to explore much richer stimulus spaces. A second related paradigm with a Gibbs sampler interpretation is described by [15], studying subjective randomness by having participants impute missing parts of coin-flip sequences. Our approach differs in soliciting continuous rather than discrete judgments. A third related paradigm is the multidimensional method of adjustment, where participants simultaneously adjust multiple

sliders to make a stimulus match a certain criterion (e.g., [31]). GSP differs from the latter in providing a principled way to share the task between participants, and a coherent probabilistic model relating slider movements to the utility function.

## 3 Studies

### 3.1 Color

Our first study concerns a particularly low-dimensional perceptual space: color. This kind of perceptual space should provide a useful sanity check for any semantic sampling procedure: if a procedure fails here, it is surely even less likely to succeed in high-dimensional perceptual spaces.[2]

We tested our sampling methods on recovering the colors associated with eight words: 'chocolate', 'cloud', 'eggshell', 'grass', 'lavender', 'lemon', 'strawberry', and 'sunset'. We parameterized color space using the perceptually oriented HSL scheme [32], where each color is encoded as three integers: hue, saturation, and lightness, taking values in [0, 360], [0, 100], and [0, 100] respectively.

The first sampling method was MCMCP, implemented with a Gaussian proposal function of standard deviation 30. The second method was standard GSP. The third method was aggregated GSP, collecting 10 slider responses for each iteration, and propagating the mean response to the next iteration. Each method was evaluated using across-participant chains of length 30, with five chains per color category, with each chain's starting location sampled from a uniform distribution over the color space (Exp. 1a, 1b, 1c). All participants ($N = 422$) were recruited from Amazon Mechanical Turk (AMT) and pre-screened with a color-blindness test and a color-vocabulary test before continuing with the online experiment (Appendix C). Each participant contributed up to 40 trials for a given method. In each case, the participant was presented with a word (e.g., 'lavender'), and asked to choose the color that best matched that word with either a binary choice interface (MCMCP) or a slider (GSP) (Fig. 1A).

There are several ways that one could evaluate the success of an MCMCP or GSP procedure. Here we follow previous work by having participants rate how well samples match the target category [12], but see Appendix C for an alternative analysis. We elicited c. 5.2 ratings per sample from a new participant group ($N = 322$, Exp. 1d); participants were presented with the target word from the original chain, and asked to judge how well the color matched this word on a scale from 1 (not at all) to 4 (very much). The results indicate a clear advantage for GSP over MCMCP, with GSP converging faster and on higher ratings; they also show that aggregation robustly improved ratings (Fig. 1B, 1C). Inspecting Fig. 1B, it is clear that many MCMCP samples poorly reflected their semantic category; meanwhile, GSP produced considerably fewer poor samples, and aggregated GSP even fewer. Investigating further, we found that the poor performance of MCMCP persisted when (a) normalizing for the longer duration of GSP trials (Fig. S8), (b) trying different proposal widths (Exp. 1e, 153 participants, Fig. S9), (c) using different questions (Exp. 1f, 190 participants, Fig. S10), (d) implementing within-experiment aggregation (Exp. 1g, 1h, 572 participants), (e) implementing post-hoc aggregation (Fig. S12), and (f) accounting for the trade-off between mode-seeking and exploration (Exp. 1i, 270 participants, Fig. S13). The implication is that, when the stimulus space is well-parameterized, GSP substantially improves sampling quality over MCMCP. In addition, it is clear that aggregation improves sampling quality still more at the cost of additional participant trials.

As an exercise, it is useful to reflect on how the stimulus space and the participants might constrain the generalizability of these results. The stimulus space presents little problem; every visible color has a close neighbor in the HSL scheme used here. However, the results should not be expected to generalize globally, given well-documented cross-cultural variations in color-naming [7].

### 3.2 Emotional prosody

This study concerns a long-standing psychological question: how the way that a sentence is spoken (its *prosody*) communicates the speaker's emotional state [33]. Prior research mostly depends on recordings of actors expressing particular emotions, but such stereotypical recordings might not fully reflect natural emotion perception [34]. GSP provides a way to study prosody perception without actors, instead generating emotional prosody directly from the perceptual judgments of listeners.

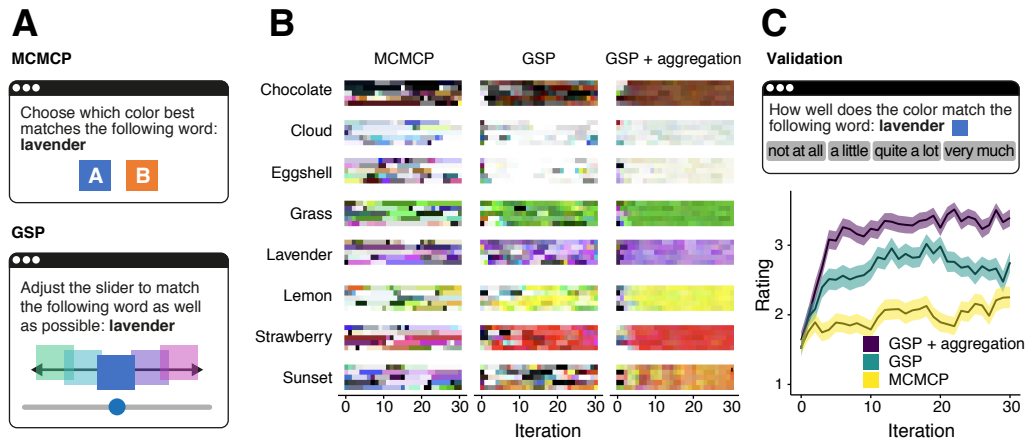

Figure 1: Sampling color representations. **A**: MCMCP/GSP instructions. **B**: Generated samples. **C**: Task and results for the validation experiment (95% confidence intervals over participants).

We began with three sentences from the Harvard sentence corpus [35] recorded by a female speaker [36], chosen to facilitate comparison with previous research; these sentences are phonologically balanced and semantically neutral. We defined our stimulus space in terms of seven parametric manipulations, corresponding to *duration* (speeding up or slowing down the fragment), *intensity variation* (rate and depth) and *pitch* (absolute level, range, slope, and F0 perturbation). We explored this space using 220 within-participant GSP chains, each comprising 21 iterations, and each beginning with the original unaltered recording (Exp. 2a). Participants ($N = 110$) were recruited from AMT, pre-screened with the audio test of [37], and each randomly assigned to either 'anger', 'happiness', or 'sadness' (Fig. 2A). Each participant contributed two chains corresponding to different sentences.

Fig. 2B plots mean feature values for the different emotional categories. Sad speech was marked by long duration, reduced pitch range, shallow pitch slope, and high F0 perturbation. Happiness had short duration, increased mean pitch, shallow pitch slope, and high pitch range. Anger had short duration, low mean pitch, falling pitch slope, and high pitch range. These characterizations are generally consistent with previous research (e.g., [38]). We also observed interesting patterns of feature correlations. For example, we found duration and F0 perturbation to be correlated for sadness ($r = .28$) but not for the other emotions (anger: $r = -.03$, happiness: $r = .00$); in contrast we found that pitch level and pitch slope were positively correlated in all three emotions (Fig. S15). This suggests a new way to explore the perceptual spaces of perceived emotions, contrasting with previous literature that mainly focuses on unique contributions of single dimensions.

We then evaluated the resulting samples with a new participant group ($N = 161$), who rated how well samples matched each emotion on a four-point scale, producing c. 5.4 ratings per stimulus (Exp. 2c). Ratings increased steadily for the first sweep of the parameter vector and then plateaued with a reliable mean contrast of 0.9 points (Fig. 2C). We also replicated the results with across- instead of within-participant chains (Exp. 2b, 2d, 210 participants, Fig. S14A).

These results imply that GSP is effective for exploring emotional prosody, and for generating emotional stimuli without the confounds of acted recordings. Nonetheless, there are clear ways in which this work could be extended. The stimulus space was defined by a limited set of manipulations, such as mean pitch, pitch slope, and F0 perturbation; this set could be extended to include for example spectral features or more granular pitch manipulations [39, 40]. The stimuli all correspond to English sentences, and the participants were all US participants; our results should not be assumed to generalize outside this cultural context [41, 42]. Moreover, all stimuli were synthesized with a female voice, so the results should not be assumed to generalize to male speakers.

### 3.3 Musical chords

Our third study concerns the subjective pleasantness of musical pitch combinations, or *chords*. For Westerners, this domain is highly multimodal, containing many prototypes of 'pleasant' (or

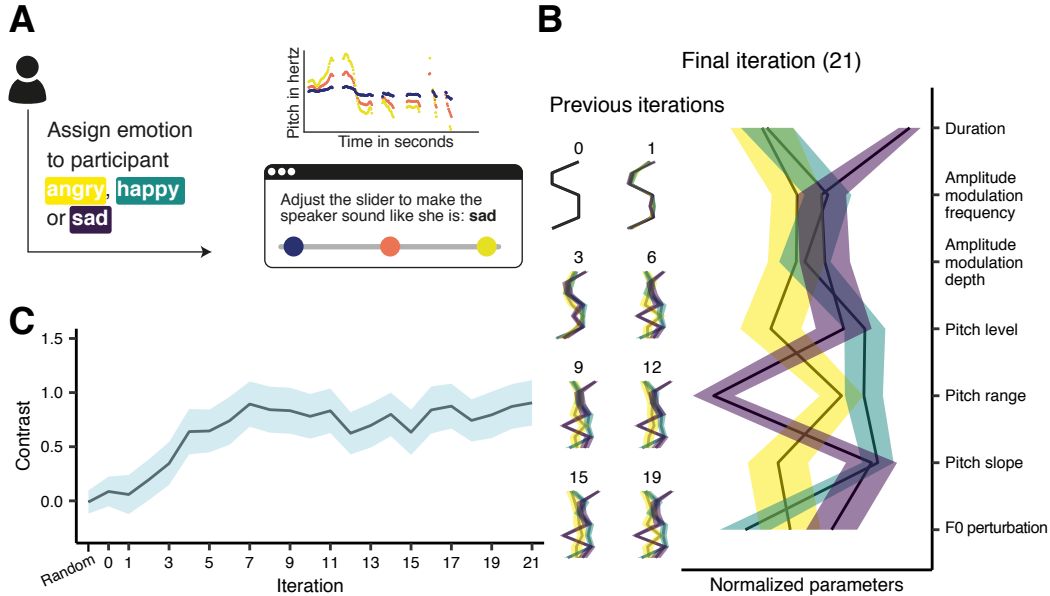

Figure 2: Sampling emotional prosody. **A**: Overview of the GSP task. **B**: Mean feature values by iteration. **C**: Mean 'contrast' ratings, corresponding to the mean rating for the 'correct' emotion minus the mean rating for the 'incorrect' emotions (95% confidence intervals over participants).

'consonant') chords. Exhaustively exploring this continuous space is difficult for conventional methods, and so far such research has been limited to single pitch intervals or to specific tuning systems [43–46]. Here we investigate whether GSP can help us to characterize the continuous space of *pairs* of pitch intervals without restricting stimuli to a given tuning system.

Our stimulus space comprised two continuous intervals, specifying the logarithmic distance from the bass tone in the range 0.5–11 [47]. The standard Western tuning system corresponds to integer coordinates in this space. We explored this space with 50 across-participant GSP chains of length 40, whose starting locations were sampled from a uniform distribution over the stimulus space. The participants ($N = 134$) were recruited from AMT and pre-screened with the audio test of [37] (Exp. 3a). These participants were instructed to make each chord as 'pleasant' as possible (Fig. 3A). In a subsequent validation experiment, participants ($N = 168$) rated pleasantness for samples from (a) the empirical distribution and (b) the top five modes of KDEs applied to raw samples from iteration 10 onwards (Exp. 3b). Each condition received 662 ratings with up to 80 ratings per participant.

Ratings increased clearly as a function of iteration, with KDE modes scoring significantly higher than raw samples. The KDEs display a rich structure that replicates and extends prior research (Fig. 3B) [43–46]. In particular, the 1D KDE shows clear integer peaks corresponding to the Western tuning system, with dips at the semitone (1) and tritone (6); the 2D KDE additionally shows peaks at various prototypical sonorities from Western music, such as the major triad (4, 7), the first inversion of the major triad (3, 8), and a dominant seventh chord with omitted third (7, 10) (see e.g., [48]; see also Fig. S18). These results imply that GSP is effective for exploring continuous musical spaces.

Our stimulus space only contained three-tone chords, but of course real music contains many different varieties of chords. Our chord tones were synthesized using artificial harmonic complex tones; though such tones are commonly used in prior research [49], real music contains many different kinds of tones, some of which have different consonance profiles [46]. Moreover, our participant group comprised mostly US and Indian participants, yet consonance perception is known to vary cross-culturally [49]. Future work should explore how our results vary as a function of these variables.

### 3.4 Faces

Our final study addresses a particularly high-dimensional domain: images of human faces. Such images would be too high-dimensional for GSP to manipulate in their raw form, so we instead

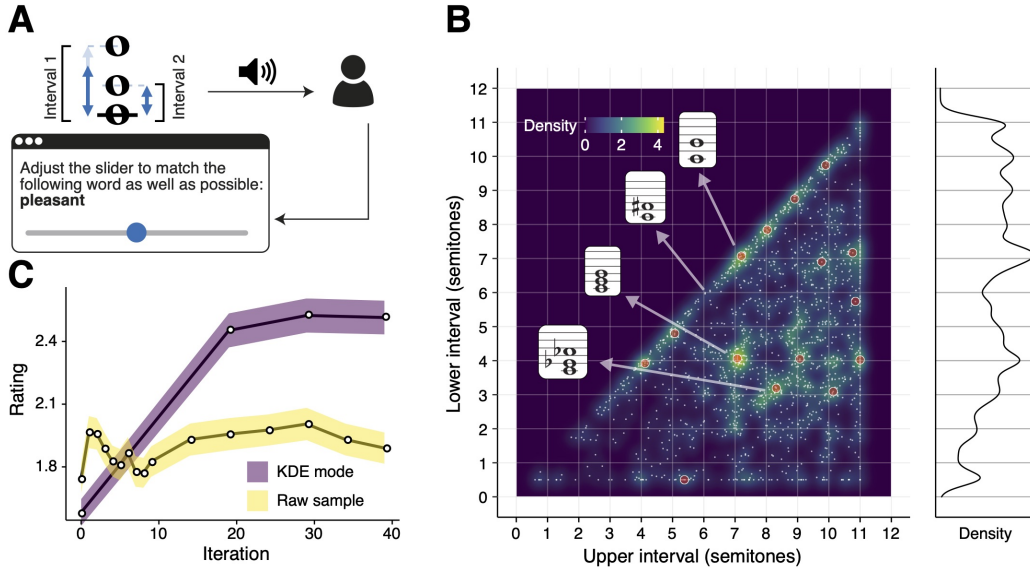

Figure 3: GSP over musical chords. **A**: Schematic illustration of the experimental task. **B**: KDE over iterations 10 to 39, with density expressed relative to a uniform distribution, the top 15 modes marked by red dots, all plotted alongside the marginal distribution of lower and upper intervals combined. **C**: Validation ratings by iteration (95% confidence intervals over responses).

parameterize the stimuli with a generative model. State-of-the-art image synthesis models typically still have high-dimensional parameter spaces, but here we build on recent work showing that the latent space of these models can be effectively navigated using principal component analysis (PCA) [50]. Following [50], we apply this approach to the generative adversarial network 'StyleGAN' [51, 52], pretrained on the FFHQ dataset of faces from Flickr [51], and applying PCA to the intermediate latent code (termed $\mathbf{w}$ in the original papers). We used the top 10 PCA components to parameterize our stimulus space, allowing these components to vary up to two standard deviations from the mean, and fixing the input latent code ($\mathbf{z}$ in the original papers) to the mean to control variability.

We used the resulting generative model to explore subjective stereotypes for the following adjectives: 'attractive', 'fun', 'intelligent', 'serious', 'trustworthy', and 'youthful', with these choices informed by prior literature (e.g., [53]). We constructed 18 across-participant GSP chains of length 50 with uniformly sampled starting locations and three chains for each adjective (Fig. 4A, Exp. 4a). We used 293 US participants from AMT, aggregating 5 trials per iteration using the arithmetic mean. We then evaluated the generated samples with a rating experiment, following the same procedure as the color experiment but collecting c. 52.1 ratings per sample from 179 US participants (Exp. 4b).

The results are illustrated in Fig. 4B–C. The GSP chains converged on highly rated samples remarkably quickly, with one full sweep of the 10 dimensions being sufficient to effectively capture the target categories as evaluated by the validation experiment. This implies that GSP can indeed successfully navigate StyleGAN's generative space. Follow-up experiments found similar success with different dimensionality reduction techniques and aggregation methods (Exp. 4c–f, Appendix F).

Samples from the GSP process will inherit certain biases from the StyleGAN model. For example, if male faces are over-represented in StyleGAN samples, they are likely to be over-represented in the GSP samples; likewise, if StyleGAN samples contain predominantly young female faces and old male faces, then GSP samples for 'youthful' are likely to be biased towards female faces. To examine such biases, we conducted a follow-up experiment analyzing the distribution of different features as subjectively rated by online participants (Exp. 4g, Appendix F). The results indicate that StyleGAN's training dataset already contains significant biases that are propagated through the modeling pipeline, and potentially contribute to the prevalence of white faces in the GSP samples, as well as gender associations for the different targets. These findings indicate the importance of interpreting GSP results in the context of their associated generative models, and of sourcing less

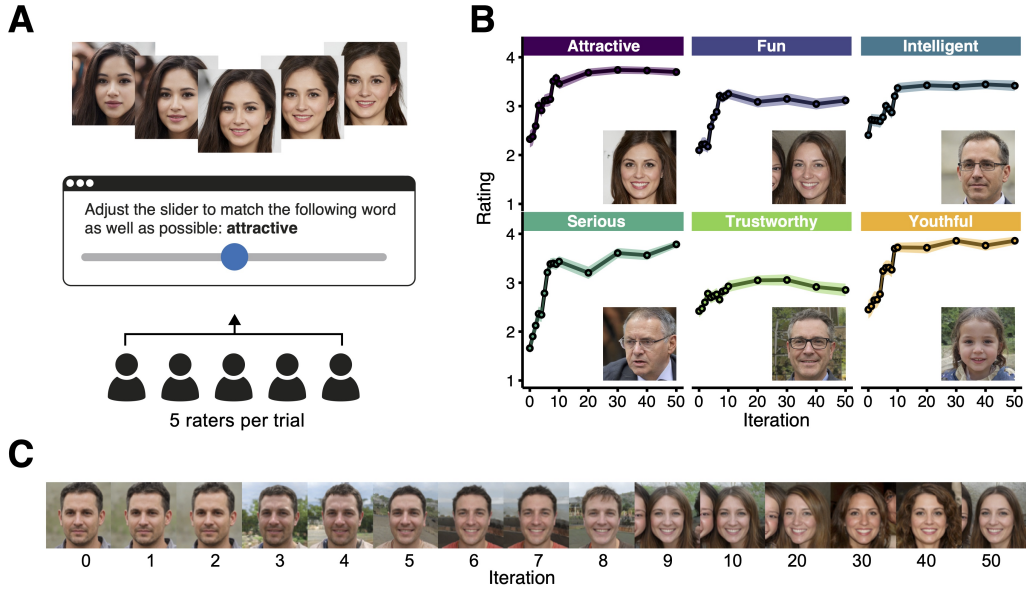

Figure 4: Sampling facial representations. **A**: Instructions for the GSP task. **B**: Results of the validation experiment, including final samples for each target adjective (95% confidence intervals over participants). **C**: Example GSP chain for 'fun', with samples ordered by iteration.

biased training datasets for future cognitive applications [54]; though StyleGAN's FFHQ dataset may be more diverse than many competing machine-learning datasets, it is clearly not bias-free. Our results will also reflect the stereotypes held by our participant group; repeating this method with different participant groups could yield interesting hypotheses concerning how facial stereotypes vary across different demographics and cultures. Appendix F describes initial experiments in this line with participant groups differentiated by gender and location (Exp. 4d, 4i, 4j).

## 4  Summary and conclusion

We have presented GSP, a new technique for extracting semantic representations from human participants. GSP organizes these participants into virtual Gibbs samplers, and thereby generates stimuli from the perceptual space associated with a given semantic representation. We have shown how this technique can recover semantic representations for a variety of perceptual domains, including color, emotional prosody, musical chords, and faces. The richness of the derived representations is compelling, and suggests many future applications in cognitive and social sciences.

GSP has several features that seem to help it converge quickly on high-quality samples. One is its continuous-slider interface, which can deliver much more information per trial than the binary choice method used by MCMCP. A second is its lack of tuning parameter, which reduces the resources required to develop a workable experiment. A third is the way in which it manipulates a single stimulus dimension at a time: it is plausible that participants find it easier to evaluate differences between stimuli when the stimuli differ on just a single perceptual dimension.

By formulating GSP and MCMCP in utility theory [19] we enable both methods to be applied to continuous as well as categorical semantic representations, while relaxing assumptions about the participant's prior and response noise. By incorporating aggregation into the conditional part of the Gibbs sampler, we increase the participant-to-stimulus ratio and thereby make GSP practical for stimuli that take a long time to generate, with the useful byproduct of averaging out perceptual noise.

The final study showed how GSP can be used to navigate the latent space of deep neural synthesis models. The important prerequisite is finding a relatively low-dimensional basis for the network for GSP to parameterize; fortunately, it seems that relatively simple techniques such as PCA can sometimes suffice for this task [50]. This approach has clear potential for helping cognitive scientists to study semantic representations in high-dimensional perceptual spaces.

## Broader Impact

This research extends the methods available to cognitive scientists who seek to characterize semantic representations in human participants. In particular, the proposed method facilitates studying much richer perceptual spaces (both in terms of dimensionality and in terms of granularity) than can be explored effectively with conventional methods.

Our research group is particularly interested in using GSP to study cross-cultural differences in perception [24, 47, 55]. In this context, exploratory techniques such as GSP are particularly useful, because they can generate valuable cognitive insights without specifying a constrained hypothesis space *a priori*. Previous work using slider interfaces with cross-cultural populations makes us relatively confident that GSP could be applied cross-culturally [56], as long as sufficient care is taken to ensure that the task is understood properly by the participants. Addressing cross-cultural populations in this way can help to ameliorate cognitive science's longstanding bias towards participants from WEIRD (Western, Educated, Industrial, Rich and Democratic) backgrounds [57].

It is important to identify potential pitfalls in applying GSP, especially when such activities have adverse ethical implications. We give three recommendations below for avoiding such mistakes.

**Do not conflate subjective judgments with objective truth.** GSP is a tool for understanding participants' subjective notions of particular semantic concepts. It does not necessarily reveal any objective truth about these concepts. This is particularly relevant in examples like our face study, where GSP is used to characterize perceived intelligence and trustworthiness. For example, GSP may suggest that participants associate glasses-wearing with intelligence: this does not mean that wearing glasses makes someone intelligent, or even that glasses wearing is necessarily associated with intelligence in the real world. Mistakes of this kind have the potential to perpetuate or amplify dangerous stereotypes in society, especially when the inferences concern race/ethnicity and gender; such an approach has a regrettable history in the now-discredited field of physiognomy. Researchers using our method and related psychological methods should be aware of this negative history, and hold their own work to a higher ethical standard to avoid causing similar harm. Consequently, GSP should not be used as tool for generating training datasets for machine-learning algorithms, or for fine-tuning the parameters or hyperparameters of such algorithms, unless the researcher makes it absolutely clear that the algorithm is being used to study human stereotypes rather than objective truths.

**Analyze, report, and ideally avoid potential biases.** Cognitive scientists must always be sensitive to potential biases in designing their stimuli and recruiting their participants. GSP is no exception to this principle. Our studies include examples of relatively simple and unbiased stimulus spaces (HSL colors; musical triads) as well as examples of relatively complex but potentially biased stimulus spaces (recordings of spoken sentences; images generated by the StyleGAN model). For practical reasons, our studies all used participants recruited from AMT; while this platform provides a relatively diverse participant group compared to the common practice of recruiting psychology students, it clearly does not represent the full diversity of the global population [58], and our results are likely to reflect culturally dependent stereotypes as a result (e.g., the Western preference for musical chords with high harmonicity, [49]). It is imperative that cognitive scientists remain vigilant concerning the potential harms of using non-diverse participant groups, both as regards making incorrect scientific conclusions and as regards perpetuating the under-representation of already marginalized parts of society [57]. We discuss these issues on a case-by-case basis above, but future cognitive work using these methods should examine these issues in greater detail. For example, we did not gather detailed personal information about our participants on variables such as race/ethnicity due to privacy reasons, but it is important that future work studying facial stereotypes takes such variables into account.

**Validate findings with rigorous hypothesis-driven experiments.** The power of combining GSP with deep generative models (e.g., StyleGAN) is that it enables the researcher to ask exploratory questions about complex naturalistic stimuli, such as 'what do people think a serious face looks like?' However, the downside of this approach is that the technique is susceptible to inheriting hidden biases from the generative model [59]. It is therefore essential that cognitive research combining GSP with deep generative models should treat the results as exploratory, and ideally validate the results with well-controlled experiments that do not rely on the generative model.

## Acknowledgments and Disclosure of Funding

The authors are grateful to David Poeppel for general help and support, and to Alec Mitchell, Jesse Snyder, Jordan Suchow, Matthew Wilkes, and Sally Kleinfeldt for their support of the Dallinger project. We would also like to thank Roya Pakzad for advising us on ethical aspects of the project. There are no conflicts of interest or external funding sources to declare.

## Footnotes

[1]Appendices, code, and raw data are hosted at `https://doi.org/10.17605/OSF.IO/RZK4S`.

[2]Additional methods, results, and demographic information for all experiments are provided in the Appendices.

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
