[Supplementary Material · gsp-appendices.pdf]

# Appendix A  Mathematical framework

In this section we present derivations of the formulas derived in the theoretical exposition of the main text. We start from a derivation of the acceptance function of MCMCP based on utility theory which we then generalize to GSP.

## A.1  MCMCP

To analyze the decision step of MCMCP, let us imagine that a participant is presented with two alternatives, $s_1$ and $s_2$, from which they are asked to choose according to some criterion $c$. In the context of utility theory, we suppose that the participant performs this task by extracting a *utility* value for each stimulus, and selecting the stimulus with the maximum utility. The utility value has two components, a deterministic component and a noise component, namely $U_i = \ell_i + n_i$, where $\ell_i$ is the deterministic utility of stimulus $i$, and $n_i$ is the associated noise. This noise component can capture intra-participant noise from sensory [1, 2] and cognitive [2, 3] processes, as well as inter-participant noise corresponding to individual differences in the utility function [4]. In the current derivation, we assume that the noises are i.i.d. and that they are Gumbel distributed (also known as type I extreme value), $n_i \sim \mathrm{Gumbel}(\mu, \gamma^{-1})$. Gumbel distributions are commonly used in discrete choice models because they approximate Gaussian noise while possessing useful analytic properties [5]. From here, the probability of choosing, say $s_1$, would be

$$p(\text{choose } s_1) = p(U_1 > U_2) = p(n_2 - n_1 < \ell_1 - \ell_2). \tag{2}$$

To proceed, we recall the following useful property of Gumbel distributions: let $X_1 \sim \mathrm{Gumbel}(\mu_1, \gamma^{-1})$ and $X_2 \sim \mathrm{Gumbel}(\mu_2, \gamma^{-1})$ be two independent variables, then the difference is logistically distributed, namely, $X_1 - X_2 \sim \mathrm{Logistic}(\mu_1 - \mu_2, \gamma^{-1})$. Thus, we see that the right hand side of (2) is simply the cumulative distribution function of the logistic distribution $\mathrm{Logistic}(0, \gamma^{-1})$, from which it follows that

$$p(\text{choose } s_1) = \frac{1}{1 + e^{-\gamma(\ell_1 - \ell_2)}} \tag{3}$$

which is the desired result (see Section 2.1 in the main paper).

## A.2  GSP

Let us now generalize the analysis of MCMCP to GSP. Recall that in the GSP step, a participant is presented with a slider that is associated with an active dimension, say $z_k$, from which they are asked to select a value. To analyze the decision step, let us discretize the slider into a set of points $\{z_k^i\}_i$, and let $z_{-k}$ denote the other fixed dimensions. Similar to the MCMCP case, we assume that the participant extracts a utility value for each stimulus along the slider, namely, $U_i = \ell(z_k^i, z_{-k}) + n_i$, with the noise being i.i.d. and Gumbel distributed $n_i \sim \mathrm{Gumbel}(\mu, \gamma^{-1})$, and we assume that they choose the alternative with the highest utility. Such a choice model is known in the literature as the multinomial logit [5]. For completeness, let us derive the formula for the probability of choosing the alternative $z_k^i$. We have

$$p(z_k^i | z_{-k}) = p\left(\bigcap_{j \neq i} U_i > U_j\right)$$

$$= \int_{-\infty}^{\infty} d\epsilon \, p\left(\bigcap_{j \neq i} n_j < \ell_i - \ell_j + n_i \,\middle|\, n_i = \epsilon\right) p(n_i = \epsilon)$$

$$= \int_{-\infty}^{\infty} d\epsilon \prod_{j \neq i} p(n_j < \ell_i - \ell_j + \epsilon) p(n_i = \epsilon)$$

$$= \gamma \int_{-\infty}^{\infty} d\epsilon \, \exp\left\{-\left(1 + \sum_{j \neq i} e^{-\gamma(\ell_i - \ell_j)}\right) e^{-\gamma(\epsilon - \mu)} - \gamma(\epsilon - \mu)\right\}$$

$$= \frac{e^{\gamma \ell_i}}{\sum_j e^{\gamma \ell_j}}$$

where in the third equality we used the fact that the noises are independent, and in the fourth equality we plugged in the standard formulas for the probability and cumulative distributions of the Gumbel distribution. The fifth equality follows from substituting $\epsilon' = \gamma(\epsilon - \mu)$ and noticing that the sum over exponentiated utility differences is a positive number, so that the integral identity $\int_{-\infty}^{\infty} dx \exp\{-\lambda e^{-x} - x\} = 1/\lambda$ holds. Thus, substituting $\ell_i = \ell(z_k^i, z_{-k})$ we arrive at the desired equation, that is, Equation (1) in the main paper. Notice also that in the case of two alternatives, this derivation recovers the acceptance function of MCMCP.

Both derivations of the MCMCP and GSP choice probabilities relied on two main assumptions regarding the nature of the noise: (a) it is i.i.d., and (b) it is Gumbel distributed. Starting from the latter, notice that the derivation of the GSP choice probability makes it clear how to generalize to other types of noise. Indeed, up to (and including) the third equality, we relied only on the i.i.d. nature of the noise. Moreover, the third equality provides a prescription on how to generalize: for a given choice of noise model, simply plug in the right cumulative function and probability distribution of that model. Thus, for a Gaussian noise for example, that is, $n_i \sim \mathcal{N}(\mu, \sigma^2)$, we have

$$p(z_k^i | z_{-k}) = \frac{1}{\sqrt{2\pi\sigma^2}} \int_{-\infty}^{\infty} d\epsilon \prod_{j \neq i} \Phi\left(\frac{\ell_i - \ell_j + \epsilon}{\sigma}\right) e^{-\frac{\epsilon^2}{2\sigma^2}} \tag{4}$$

where $\Phi$ is the normal cumulative distribution function. This is known as the independent probit model [5]. Of course, unlike the Gumbel case, in the Gaussian case this does not result in a closed form formula. This, however, does not prevent the GSP process from exploring the utility terrain of the model, given the functional similarity between Gumbel and normal distributions.

If the noise cannot be assumed to be i.i.d., the third equality no longer holds. We see two main potential sources of i.i.d. violations in this paradigm:

1. Intra-participant correlation (different participants have different utility functions);
2. Neighboring-point correlation (neighboring points on the slider are likely to receive correlated noise).

Intra-participant correlation has different implications for across- and within-participant chains (Fig. S1). In across-participant chains, each participant only contributes one observation to the chain, so intra-participant correlation never manifests. In within-participant chains, all observations in a given chain come from the same participant, meaning that the i.i.d. assumption remains unviolated, and each chain ends up approximating the underlying utility function for the individual participant. The population-level utility function can then be approximated by aggregating over chains.

Neighboring-point correlation could have a subtle effect on the derivations presented here. Future work could revise our model to include a correlation structure for these points, for example following the correlated multinomial probit model where noise values are taken from a joint Gaussian distribution with a specified correlation structure [5].

Our derivation also does not take into account context effects, whereby the participant's previous trials influence their responses to the present trial. In particular, it is possible that the utility value

Figure S1: Illustration of different chain designs.

Figure S2: Computational infrastructure used for data collection.

for a given stimulus changes when the participant has already experienced that stimulus multiple times. This possibility is particularly high in within-participant chains, where the same participant experiences many stimuli from adjacent steps in the Gibbs sampler; in contrast, across-participant chains mostly avoid this effect by preventing the participant from experiencing multiple stimuli from the same chain (Fig. S1). We tested the strength of this effect in the emotional prosody experiment, conducting both a within-participants and an across-participants version of the same paradigm. We found that the results did not differ materially between the two, implying that memory effects were not a significant problem for this paradigm.

The above GSP derivation also assumes that the participant visits all slider positions. If this assumption is violated, the denominator in the GSP choice probability would cover only a subset of the locations, effectively reducing the granularity of the slider. We can try and minimize this effect experimentally, by forcing the participant to explore a certain amount of the stimulus space before proceeding to the next trial, but it is often impractical to enforce exhaustive exploration. However, we expect that the consequences of this assumption violation are not severe for two reasons: (a) participants tend to focus on the parts of the slider that contain most of the utility/probability mass, and (b) participants can extrapolate between slider locations to estimate the utility values of intermediate points. Nonetheless, we would like to explore this assumption more in future work.

## Appendix B   General methods

### B.1   Implementation

We implemented all experiments in PsyNet, our under-development framework for implementing complex experiment paradigms such as GSP and MCMCP. This framework builds on the Dallinger platform for experiment hosting and deployment.[3] Participants interact with the experiment via their web browser, which communicates with a back-end Python server cluster responsible for organizing the timeline of the experiment (Fig. S2). This cluster is mostly managed by Heroku,[4] and comprises a customizable collection of virtual instances that share the experiment management and stimulus generation workload, as well as an encrypted Postgres database instance for storing results. In some experiments we additionally used Amazon Web Services (AWS) S3 storage for hosting stimuli, and an AWS Elasic Compute Cloud (EC2) instance with an NVIDIA K80 GPU for deep neural network synthesis.[5] Code for the implemented experiments can be found at `https://doi.org/10.17605/OSF.IO/RZK4S`.

## B.2 Participants

All participants provided informed consent in accordance with the Max Planck Society Ethics Council approved protocol (application 2018-38). All participants were recruited from Amazon Mechanical Turk (AMT),[6] which is an online service for crowd-sourcing workers for online tasks. Here the only universal constraint we placed on recruitment was that participants must be at least 18 years of age, and have a 95% or higher approval rate on previous tasks on AMT; this approval criterion is meant to help recruit reliable participants. In some experiments we also constrained the worker to be a US resident.

When designing the experiment, each component was given an estimate for the average time it should take to complete; participants were then paid at a US $9/hour rate according to how much of the experiment they completed. Importantly, participants were still paid a proportional amount even if they left the experiment early on account of failing a pre-screening task.

A total of 5,178 participants took part in the 25 experiments reported in this paper, excluding those who failed pre-screening tests. For the participants who reported demographic information, self-reported ages ranged from 18 to 89 ($M = 35.25$, $SD = 10.37$), and 35.74% identified as female (63.9% male and 0.36% other).[7]

These participants may be further differentiated into two groups: those who participated in the main experiments and those who participated in the validation experiments. These two groups had similar compositions, with the main participant group (Table S1) comprising 2,967 participants (35.95% female, 63.64% male, 0.41% other; ages 18–89, $M = 35.25$, $SD = 10.42$), and the validation group (Table S2) comprising 2,211 participants (35.22% female, 64.53% male, 0.25% other; ages 18–74, $M = 35.25$, $SD = 10.24$).

The musical chord study also collected additional information about musical expertise. Participants in the main chord experiment reported 0–25 ($Med = 2$, $M = 4.26$, $SD = 6.23$) years of musical experience (i.e., playing an instrument or singing), whereas participants in the corresponding validation experiment reported 0–64 ($Med = 2$, $M = 4.39$, $SD = 7.57$) years of musical experience.

Participant recruitment was managed by PsyNet. For the across-participant chain experiments, we specified a desired number of chains and a desired length for these chains, and participants were then automatically recruited until the chains reached their desired lengths. For the within-participant chain experiments, we specified a desired number of completed participant sessions, and recruitment continued until this threshold was met. For the rating experiments, we chose a desired number of ratings per experimental condition[8] such that we expected that any variation in the resulting condition means should primarily reflect the stochasticity of the original sampler rather the stochasticity of participant raters. As a rule of thumb, we aimed for approximately 150 participants per validation study, scaling this number accordingly when the validation study compared multiple methods. Participants were then automatically recruited until the minimum number of ratings per experimental condition was reached.

Table S1: Main experiments

| Experiment | Method | Rep. | Dim. | Iter. | Agg. | Chain type | $N$ | Pre-screening | US-only | Validated in |
|---|---|---|---|---|---|---|---|---|---|---|
| 1a Color (MCMCP) | MCMCP | 8 | 3 | 30 | 1 | Across | 57 | CB, CV | No | Exp. 1d, 1h |
| 1b Color (GSP) | GSP | 8 | 3 | 30 | 1 | Across | 53 | CB, CV | No | Exp. 1d, 1h |
| 1c Color (agg. GSP) | GSP | 8 | 3 | 30 | 10 | Across | 312 | CB, CV | No | Exp. 1d, 1h |
| 1e Color (MCMCP proposal) | MCMCP | 8 | 3 | 30 | 1 | Across | 153 | CB, CV | No | - |
| 1f Color (questions) | GSP/MCMCP | 8 | 3 | 30 | 1 | Across | 190 | CB, CV | No | - |
| 1g Color (agg. MCMCP) | MCMCP | 8 | 3 | 30 | 10 | Across | 302 | CB, CV | No | Exp. 1h |
| 2a Prosody (within) | GSP | 3 | 7 | 21 | 1 | Within | 110 | Audio | Yes | Exp. 2c, 2d |
| 2b Prosody (across) | GSP | 3 | 7 | 20 | 1 | Across | 57 | Audio | Yes | Exp. 2d |
| 3a Musical chords | GSP | 1 | 2 | 40 | 1 | Across | 134 | Audio | No | Exp. 3b |
| 4a Faces | GSP | 6 | 10 | 50 | 5 | Across | 293 | CV | Yes | Exp. 4b, 4d, 4g |
| 4c Faces (KDE) | GSP | 6 | 10 | 50 | 5 | Across | 278 | CV | Yes | Exp. 4d |
| 4e Faces (basis) | GSP | 1 | 10 | 30 | 5 | Across | 167 | CV | Yes | Exp. 4f |
| 4h Faces (art) | GSP | 6 | 10 | 50 | 5 | Across | 260 | CV | Yes | - |
| 4i Faces (global, KDE) | GSP | 6 | 10 | 50 | 5 | Across | 269 | CV | No | Exp. 4d |
| 4j Faces (dating) | GSP | 1 | 10 | 30 | 5 | Across | 332 | CV | Yes | - |

*Note.* 'Rep.' indicates the number of semantic representations that were tested; 'Dim.' indicates the dimensionality of the stimulus space; 'Iter.' indicates the number of iterations in each chain; 'Agg.' indicates how many participants contributed to each iteration of the GSP chain; '$N$' denotes the number of participants included in the final analysis; 'CB' denotes the color blindness pre-screening task; 'CV' denotes the color vocabulary pre-screening task; 'US-only' indicates whether the participant group was restricted to US residents; 'Exp.' denotes 'Experiment'.

Table S2: Validation experiments

| Experiment | Ratings per participant | Ratings per stimulus | Total stimuli | $N$ | Pre-screening | US-only | Validating |
|---|---|---|---|---|---|---|---|
| 1d Color (original) | 60 | 5.2 | 3,720 | 322 | CB, CV | No | Exp. 1a, 1b, 1c |
| 1h Color (inc. agg. MCMCP) | 60 | 3.3 | 4,960 | 270 | CB, CV | No | Exp. 1a, 1b, 1c, 1g |
| 1i Color (uniform sample) | 60 | 4.2 | 4,000 | 280 | CB, CV | No | Exp. 1a, 1b, 1c, 1g |
| 2c Prosody | 147 | 5.4 | 4,383 | 161 | Audio | Yes | Exp. 2a |
| 2d Prosody | 132 | 4.1 | 4,874 | 153 | Audio | Yes | Exp. 2a, 2b |
| 3b Musical chords | 80 | 16.4 | 820 | 168 | Audio | Yes | Exp. 3a |
| 4b Faces (original) | 80 | 52.1 | 275 | 179 | CV | Yes | Exp. 4a |
| 4d Faces (aggregation, location) | 80 | 25.6 | 815 | 261 | CV | No | Exp. 4a, 4c, 4i |
| 4f Faces (basis) | 59.9 | 4.3 | 260 | 131 | CV | No | Exp. 4e |
| 4g Faces (bias) | 78.9 | 3.2 | 7,056 | 286 | CV | Yes | Exp. 4a |

*Note.* '$N$' denotes the number of participants included in the analysis; 'CB' denotes the color blindness pre-screening task; 'CV' denotes the color vocabulary pre-screening task; 'US-only' indicates whether the participant group was restricted to US residents; 'Exp.' denotes 'Experiment'. In the row corresponding to Exp. 4g, the number of stimuli (7,056) corresponds to 7 (the number of questions) multiplied by 1,008 (the number of images).

Write down the number in the image.

Figure S3: Example trial from the color blindness pre-screening task.

## B.3  Pre-screening tests

A useful technique for improving the quality of data from online participants is to implement pre-screening tests designed to screen out participants likely to deliver low-quality data [6]. Here we used three pre-screening tests in various combinations: a color blindness test, a color vocabulary test, and an audio test. These tests are primarily intended to screen out participants who do not meet certain explicit criteria such as wearing headphones, but they also help to screen out participants with a minimal degree of English comprehension, or automated scripts ('bots') masquerading as participants [7].

The color blindness test was derived from the well-known Ishihara color blindness test [8]. Here participants had to respond to six trials where the task was to transcribe a number from an image, with the contrast of the image being designed such that it is difficult to perform if the participant suffers from color perception deficiencies (see Fig. S3 for an example trial). The image was set to disappear after three seconds to encourage quick responses. Each participant had to take six such trials; to pass, they had to answer at least four of these six trials correctly.

The color vocabulary test was constructed by taking six English color words that require a relatively good vocabulary knowledge to understand: 'turquoise', 'magenta', 'granite', 'ivory', 'maroon', and 'navy'. None of these words were used in the other experiments. We associated each word with an RGB definition sourced from Wikipedia, and presented the participant with six trials where they were presented with a color and had to choose which of the six words corresponded to that color (see Fig. S4). The pass threshold was a score of four out of six.

The audio pre-screening task, originally developed in [6], was intended to ensure that participants were wearing headphones and could hear perceive subtle sound differences. The task has participants perform a three-alternative forced-choice task to identify the quietest of three tones. These tones are constructed to elicit a phase cancellation effect, such that when played on loudspeakers the order of quietness changes, causing the participant to respond incorrectly. Each participant had to take six such trials; to pass, they had to answer at least four of these six trials correctly.

## B.4  Performance incentives

In order to further improve data quality, some of our experiments (specifically, all but the emotional prosody experiments) additionally included a financial incentive for participants to provide high-quality data. Prior to the main part of the experiment, we informed all participants of this incentive, using the following text:

Figure S4: Example trial from the color vocabulary pre-screening task.

Figure S5: Example trial from the audio pre-screening task.

> The quality of your responses will be automatically monitored, and you will receive
> a bonus at the end of the experiment in proportion to your quality score. The best
> way to achieve a high score is to concentrate and give each trial your best attempt.

We purposefully left the definition of 'quality' vague, so as to avoid encouraging participants to 'game' a particular aspect of response quality. Of course, our tasks were subjective, and so there was no meaningful way to define a high-quality answer *a priori*. Instead, our approach was to use consistency as a proxy for quality; the rationale is that a participant who takes the task seriously and carefully is likely to deliver consistent responses when administered the same trial multiple times, in contrast to a participant who does not pay attention to the task and simply answers randomly.

We estimated consistency as follows. Once a participant finished all of their 'main' experiment trials, they then received a small number (4–8, depending on the experiment) of trials that repeated randomly selected trials from the earlier part of the experiment. The data from these trials contributed solely to consistency estimation, not to chain construction. In GSP trials and four-point rating trials, consistency was quantified by taking the Spearman correlation between the two sets of answers; for MCMCP trials, consistency was quantified by taking the percentage agreement between the two sets of answers. Participants were then given a small monetary bonus in proportion to the resulting consistency score, ranging from zero dollars for chance performance up to one dollar for perfectly consistent performance.

## B.5 Chain construction

In all experiments except the emotional prosody experiment, we randomized the starting locations of each chain by randomly sampling from a uniform distribution over the range of permissible feature values. In the case of emotional prosody, we found this randomization problematic because it often led to unrealistic parts of the stimulus space. In this case we therefore initialized each chain at a 'null' state corresponding to the unaltered reference sentence.

In a given trial of a GSP experiment, the participant's slider manipulated exactly one dimension of the stimulus. To counteract any potential biases towards left or right slider directions, we randomized the effective direction of the slider on each trial, such that approximately half of the time the right of the slider corresponded to positive feature values, and the other half of the time it corresponded to negative feature values.

Our experiments implemented both within-participant and across-participant chains (Fig. S1). In within-participant chains, the entire chain is completed by just one participant, and the resulting samples reflect the semantic representations of that single participant. In across-participant chains, each iteration comes from a different participant, and the samples then reflect shared semantic representations across participants.

Across-participant chains are more complex to implement because of the interaction between multiple participants. Each time a participant is ready to take a new trial, it is necessary to scan the different chains in the experiment and identify one that satisfies the following conditions:

1. The chain is not full (i.e., it has not reached its specified quota of iterations);
2. The participant has not already participated in that chain;
3. No other participants have been assigned to that particular iteration of the chain.

The last point – ensuring that multiple participants are not assigned to the same iteration of a chain – is important for the efficiency of data collection, but it can cause problems when a participant claims a particular iteration of the chain and then drops out of the experiment, potentially blocking any future additions to that chain. We therefore implemented a time-out parameter for this experiment, set to 60 seconds, after which the participant's pending trial was invalidated and the chain was unblocked.

In within-participant chains we are free to discard all of a participant's data when they drop out of an experiment partway through. This is not practical in across-participant chains, however, where many subsequent participants might have built on the data previously contributed by this participant. In the latter case, we therefore retain the participant's contributions even when they drop out of the experiment.

## Appendix C   Color

### C.1   Supplementary methods

We chose eight words designed to be moderately but not overly familiar to English speakers that we anticipated to evoke strong color associations. These words were 'chocolate', 'cloud', 'eggshell', 'grass', 'lavender', 'lemon', 'strawberry', and 'sunset'. We then explored the perceptual spaces associated with these eight words using GSP and MCMCP.

We implemented GSP and MCMCP using the Hue, Saturation, Lightness (HSL) color space. We chose this color space over the Red, Green, Blue (RGB) color space because it is generally considered to better reflect how humans perceive color relationships. In this space, each color is encoded as three integers: hue, saturation, and lightness, taking values in $[0, 360)$, $[0, 100]$, and $[0, 100]$ respectively.

MCMCP relies on the specification of a proposal function. In our main experiment, we used a Gaussian distribution with a standard deviation of 30, chosen on the basis of internal piloting, and rounding samples to the nearest integer values. In MCMCP the proposal distribution should be symmetric, which can be problematic to satisfy when the sampler reaches the boundaries of the sample space. We addressed this problem by computing the proposal distribution modulo the scale range, such that moving past the top of the scale means returning to the bottom of the scale. This works particularly well for hue, which is already defined as a circular space, with both 0 and 360 corresponding to the color red. It works less well for saturation and lightness, because these are linear scales; however, as we chose target colors occupying central regions of these two scales, we expected that these boundary effects should not materially influence MCMCP's performance.

We implemented simple web interfaces for the MCMCP and GSP tasks. In the MCMCP task, participants were presented with pairs of colors, and had to choose which color best represented a target word (Fig. S6). In the GSP task, participants were presented with a single color that constantly updated to reflect the current position of a slider; participants were then instructed to move the slider to make the color represent a target word as well as possible (Fig. S7).

Table S3: Questions used in Exp. 1f.

| Label | MCMCP | GSP |
|---|---|---|
| Probability | Choose which colour is most likely to come from the following category. | Adjust the slider to make the color as likely as possible to come from the following category. |
| Goodness | Choose which colour best matches the following word. | Adjust the slider to match the following word as well as possible. |
| Typicality | Choose which colour is most typical of the following category. | Adjust the slider to make the color as typical as possible for the following category. |

*Note.* All color experiments except for Exp. 1f used only the 'Goodness' question; Exp. 1f tested all three questions.

For each given sampling method we constructed five across-participants chains per adjective, yielding 40 chains in total. Each chain was filled to a length of 30 states, not including the initial random state. Each participant contributed a maximum of 40 trials to the chains for a given sampling method (Exp. 1a–c; see Table S1 for participant numbers).

The aggregated GSP experiment combined 10 trials for each iteration of the Gibbs sampler (Exp. 1c). These 10 trials were combined using the arithmetic mean in the case of saturation and lightness, and the circular mean in the case of hue. These means were then propagated to the next iteration of the Gibbs sampler.

We also ran five follow-up experiments to better understand the relative performance of GSP and MCMCP (see also Tables S1 and S2):

- We tested MCMCP with five different proposal function standard deviations: 10, 20, 30, 40, and 50, all expressed on the integer color scale (Exp. 1e).

- We reran the MCMCP and GSP experiments with three different kinds of questions designed to probe different notions of utility and category membership (Table S3, Exp. 1f).

- We reran the MCMCP experiment using 10-fold aggregation (Exp. 1g), and validated it alongside the other methods (Exp. 1h).

- We tested 4,000 colors randomly sampled from a uniform distribution over the HSL space using the same rating procedure as Exp. 1d (Exp. 1i).

The validation experiment (Exp. 1d) used the same pre-screening procedure as the chain-construction experiments. A minimum of five ratings were collected for each sample generated in the former experiments, with the constraint that participants could not rate the same stimulus more than once. Participants were assigned pseudo-randomly to stimuli such that the number of ratings accumulated evenly for each stimulus. In each trial, participants were presented with the target word from the original chain, and asked to judge how well the colour matched this word on a scale from 1 (not at all) to 4 (very much). A given participant's ratings were only included in the final tallies if they completed the entire validation experiment. See Table S2 for participant numbers.

## C.2 Supplementary results

In the main paper we identified a clear advantage for GSP over MCMCP, given chains of the same length and the same amount of aggregation. However, we were concerned about several possible confounds, which we will now discuss alongside corresponding analyses.

*Claim:* **GSP trials are more time-consuming than MCMCP trials. Even if GSP requires fewer trials to achieve good sample quality, if these trials take much longer, then GSP will end up being practically slower than MCMCP.** Fig. S8 plots validation ratings for the different sampling methods, with the horizontal axis now corresponding to the total participant time invested in the respective chains (Exp. 1a–c). We estimated total participant time by taking iteration number and multiplying it by the median participant time spent on the two different trial types. The results indicate that non-aggregated GSP still clearly outperformed MCMCP despite the longer duration

Figure S6: Screenshot from the color MCMCP implementation.

Figure S7: Screenshot from the color GSP implementation.

Figure S8: Mean sample ratings as a function of the participant time invested in chain construction (Exp. 1a–c, 1d), with time plotted on a log scale (95% confidence intervals over participants).

of its individual trials. It is difficult to make a clear statement about the relative performance of aggregated GSP because its profile overlaps minimally with the other two methods; however, the figure implies that non-aggregated GSP outperforms aggregated GSP for the first few iterations, with aggregated GSP then overtaking at a later point. This is consistent with our expectations: the fast-but-noisy non-aggregated GSP can quickly escape its low-probability starting states, but the same noise prevents it from converging as precisely as aggregated GSP in later iterations.

*Claim:* **MCMCP has a tuning parameter corresponding to the width of the proposal function. Perhaps the relatively poor performance of MCMCP was simply due to the wrong choice of proposal width.** We tabulated samples from the previously described control experiment with different MCMCP proposal widths (Fig. S9, Exp. 1d). There appears to be little difference in sample quality for the different proposal widths. As expected, we see that the MCMCP chains with the smallest proposal width (10) only make local adjustments to the color, meaning that once the chain gets close to an appropriate color category, it can be carefully tweaked to resemble this category as well as possible. However, these narrow-proposal chains often fail to approach the appropriate color category in the first place, even after 30 iterations. In contrast, the wide-proposal chains explore the color space quickly, but are unable to make subtle adjustments to match specific categories. The moderate proposal width of 30 provides some compromise between these two behaviors, and seems to be a sensible choice for the MCMCP-GSP comparison.

*Claim:* **We altered the MCMCP question somewhat to better represent the notion of continuous utility as opposed to category membership. Perhaps this alteration diminished the efficacy of MCMCP in practice.** We tabulated samples from Exp. 1f which trialled different types of questions for the MCMCP and GSP tasks (Table S3, Fig. S10). Visually inspecting these plots, we struggled to discern any systematic effect of question type on the sample distributions. We do not doubt that subtle differences could be distinguished with the right kind of experiment, but it seems that in practice any such effects are small.

*Claim:* **We only evaluated MCMCP without aggregation; perhaps MCMCP with aggregation would perform as well as GSP.** We compared validation ratings for aggregated MCMCP against ratings for aggregated GSP, non-aggregated GSP, and non-aggregated MCMCP (Fig. S11). Aggre-

Figure S9: Raw color samples for MCMCP with five different standard deviations for the Gaussian proposal function: 10, 20, 30, 40, and 50 (Exp. 1e).

gated MCMCP does outperform non-aggregated MCMCP, but the difference is small compared to the difference between GSP and aggregated GSP. This makes intuitive sense: while aggregated GSP can produce very precise updates at each iteration, aggregated MCMCP can only provide one bit of information at each iteration, placing a fundamental limit on its convergence rate.

*Claim:* **A common analysis approach with MCMCP is to generate category prototypes by averaging over many samples. Perhaps MCMCP performs better when using this analysis method.** We recomputed the samples generated by the three methods using instead an incremental aggregation process, generating a summary sample for each iteration and target word by averaging all previous samples from all chains for that word, with iterations 1–6 treated as burn-in samples and hence discarded. The resulting samples are displayed in Fig. S12. The aggregation process clearly improves sample quality for MCMCP and GSP (non-aggregated), but it does not fully solve MCMCP's problem with poor sample quality. Though we do not have participant rating data for these aggregated samples, it is clear that MCMCP failed to converge on appropriate colors for chocolate, eggshell, and lavender. One might further criticize the lavender samples for being too red, the strawberry and sunset samples for not being red enough, and the lemon samples for being not yellow enough. It seems apparent that aggregating over trials does not necessarily resolve the performance issues of MCMCP in this paradigm.

*Claim:* **Our original evaluation rewards methods that produce highly prototypical category exemplars; using our utility function metaphor, one might say that the evaluation rewards mode-seeking behavior. However, there is a trade-off between mode seeking and exploration; perhaps GSP is better at mode seeking, but MCMCP is better at exploration.** We estimated a benchmark utility distribution over the stimulus space for each target word using a large-scale rating experiment (Exp. 1i), and then compared the results to the utility distributions estimated by MCMCP and GSP. To provide a visual intuition for the differences between techniques, Fig. S13 plots marginal distributions for hue as estimated by the rating, MCMCP, and GSP experiments, using a generalized additive model for the ratings and a kernel density estimator (KDE) for the MCMCP and GSP distributions, and again treating iterations 1–6 as burn-in samples. Only 'grass', 'lavender', 'lemon', and 'strawberry' are plotted here, because these are the four words with the most interpretable marginals for hue (the remaining adjectives have many very dark or very light samples, in which case differences in hue become imperceptible). From comparing GSP and MCMCP to the ratings, it is apparent that the poor performance of MCMCP is not simply due to having broader peaks, but rather comes from mislocated secondary peaks, for example red for 'grass', orange for 'lavender', blue for 'lemon', and so on. Visually inspecting the raw samples in Fig. 1B supports this impression: many of the MCMCP samples seem to be unrelated to the target category. Incidentally, the figure also helps for visualizing the effect of aggregation; we see how aggregation sharpens the

**MCMCP**

**GSP**

Figure S10: Raw color samples for MCMCP and GSP with three different kinds of questions as described in Table S3 (Exp. 1f).

Figure S11: Validation results for non-aggregated and aggregated GSP and MCMCP (Exp. 1a, 1b, 1c, 1g, and 1h). The shaded regions indicate 95% confidence intervals over participants.

Figure S12: Colors derived by averaging raw samples from iteration 7 onwards for the different sampling methods (Exp. 1a–c).

GSP peaks to clear unimodal distributions, but fails to provide much improvement for MCMCP. Future work should investigate these differences more systematically using quantitative assessments of multidimensional distribution similarity, and unpacking the potentially non-trivial relationship between sample ratings and utility values.

To summarize, it seems that none of these six considerations impact substantially on the main conclusion that GSP outperforms MCMCP for this color estimation task. Nonetheless, each of these issues could certainly be explored in more detail in future work; each perceptual domain is different, and in some cases MCMCP may become the preferred tool as a result.

## Appendix D  Emotional prosody

### D.1  Stimuli

The stimuli were created on the basis of three sentences from the Harvard sentences [9] recorded by a female speaker [10]. These sentences are phonologically balanced and semantically neutral. The stimulus space was then defined through seven continuous acoustic manipulations performed to these sentences. The manipulations were performed using the software Praat [11] and the Python package Parselmouth [12]. Pitch (F0 contour) was extracted using a pitch floor of 100 Hz and ceiling of 500 Hz (default window size) using the command `To Pitch` in Parselmouth. Before proceeding we confirmed that all contours were free of any octave jumps. From the `Sound` and the `Pitch` object, we created a `Manipulation` object using the command `To Manipulation`. From the `Pitch` object we extracted the glottal pulses using `To PointProcess`. The manipulations were then performed in the following order:

1. *Pitch level*, shifting the pitch contour by a value in the range $[-37, 37]$ Hz.

2. *Pitch range*, scaling the original pitch range (expressed in Hz) by a value in the range [20, 180]%, using the middle of the original pitch range as the center of the scaling operation.

3. *Pitch slope*, altering the original sentence's pitch slope by a value in the range $[-37, 37]$ Hz. In our case, the reference sentences always began with a falling slope, and our manipulation was never severe enough to change them to a rising slope. Instead, a positive value of our

Figure S13: Utility distributions as estimated by rating, MCMCP, and GSP experiments, treating iterations 1–6 as burn-in samples (Exp. 1a, 1b, 1c, 1g, 1i).

pitch slope feature indicates a flattened contour, and a negative value indicates a steeply falling contour.

We manipulated pitch slope in the following way. We extracted the time of the first ($t_0$) and last ($t_1$) pitch values (ignoring unvoiced segments), and then edited the pitch contour by adding the following linear function $f(t)$ to each pitch:

$$f(t) = x * \frac{t - t_0}{t_1 - t_0} \tag{5}$$

where $t$ denotes the time of the point being edited and $x$ denotes the feature value, ranging between $-37$ Hz and $37$ Hz.

We achieved this by creating an empty `PitchTier` object and populating it with the new contour using the command `Add point`. Finally we replace the old `PitchTier` in the `Manipulation` object with the new one using `Replace pitch tier`.

4. *F0 perturbation* is commonly measured as local frequency variation in the F0 contour (jitter), and corresponds approximately to the perceptual impression of hoarseness [13]. We modified F0 perturbation by converting the `PointProcess` object (representing the glottal pulses) to a Praat `Matrix` object (representing the time points of the pulses) using `To Matrix`. We changed the position of the pulses by applying the Praat formula `self + randomGauss(0, r)` where `r` was a number between 0 and 0.0001 determining the strength of the perturbation. The `Matrix` was converted back to a `PointProcess` with `To PointProcess`, and the glottal pulses in the `Manipulation` replaced using `Replace pulses`. This follows the algorithm proposed in [14].

5. *Duration*, allowed to change linearly from 80% to 120% from the original duration. To manipulate the duration we created an empty `DurationTier` object using the command `Create DurationTier`. At time 0 we placed a point with the duration value using the command `Add point 0 scalar`. We then ran `Replace duration tier` to apply the changes. Note that changing the duration did not affect the overall pitch.

6. *Intensity variation*, corresponding to a periodic amplitude modulation of the signal. This manipulation was characterized by two parameters which constituted two independent dimensions of the stimulus space: *amplitude modulation frequency* (ranging from 0–5 Hz) and *amplitude modulation depth* (ranging from 0.01–10 dB). We implemented this using the operation 'Vibrato and tremolo' as defined in [14] and implemented in Parselmouth.

## D.2  Procedure

The main chain-construction experiment (Exp. 2a) assigned each participant to one of three different emotions: happiness, sadness and anger. To ensure that all participants were familiar with the emotional concept we presented contexts that has been used in previous studies on emotional prosody [15, 16]:

> **Anger:** Please think of a situation where you experienced a demeaning offense against you and yours. For example, somebody behaves rudely toward you and hinders you from achieving a valued goal. The situation is unexpected and unpleasant, but you have the power to retaliate.

> **Happiness:** Please think of a situation where you made reasonable progress toward the realization of a goal. For example, you have succeeded in achieving a valued goal. Your success may be due to your own actions, or somebody else's, but the situation is pleasant and you feel active and in control.

> **Sadness:** Please think of a situation where you experienced an irrevocable loss. For example, you lose someone or something very valuable to you, and you have no way of getting back that what you want.

Each participant was randomly assigned to a different emotion (happy, sad, angry). After the headphone-screening task and a short demographic questionnaire, they took a practice trial to familiarize themselves with the slider. They then completed two within-participant chains corresponding to three alterations of each of the seven dimensions, alternating between both chains until both were

complete. Each chain was initialized with the feature values of the reference sentence. In each trial the participant could chose from 25 stimuli synthesized from 25 equidistant points on the slider.

In the validation experiment (Exp. 2c), each participant rated stimuli for the three emotion words (happiness, sadness, anger) in three corresponding randomly ordered blocks. Each block contained 49 stimuli, which came in four types: (a) raw samples from the GSP chains, (b) samples derived by averaging the last three iterations of the GSP chains, (c) the initial unchanged sentences, (d) samples corresponding to random feature values. Participants were presented with the same emotional contexts as the participants in the chain-construction experiment, and responded using the same four-point scale as the other experiments ('1. Not at all', '2. A little', '3. Quite a lot', '4. Very much').

We also conducted a control experiment where we switched from within-participant chains to across-participant chains, reducing the number of participants by approximately half because the original experiment proved to have more than sufficient power, and leaving all other experiment parameters unchanged (Exp. 2b). Note that reducing the number of participants should not bias the validation ratings, which only used raw samples rather than samples created by aggregating over participants. Due to a minor implementation error, this experiment only constructed chains of length 20 rather than of length 21.

In the subsequent validation component (Exp. 2d), participants rated three blocks of 44 stimuli: 20 samples from the original within-participant chains, 20 samples from the new across-participant chains, 3 random samples, and one initial unchanged sentence. In all other regards this second validation was identical to the first validation.

### D.3 Supplementary results

Fig. S14A shows the results of the within- and across-participant comparison (Exp. 2a, 2b). The resulting feature values are broadly similar between these two experiments, suggesting that memory effects did not substantively contaminate our within-participant chains. This conclusion is supported by the validation experiment, which shows similar contrast scores for both within- and across-participant chains (Fig. S14B, Exp. 2d).

Fig. S14C shows how mean feature values develop over the course of the within-participant experiment (Exp. 2a). Here we can see how most of the development of the feature values occurs over the first sweep of the feature vector (iterations 1–7), after which point the feature values stay broadly similar. Three audio examples from the same sentences in the final iteration of this experiment can be found at `https://doi.org/10.17605/OSF.IO/RZK4S` in the folder `sound-examples-prosody` with the filenames `sad|happy|angry_final_sentence.wav`, alongside for reference the initial stimulus `original_sentence.wav`.

GSP also allows us to investigate higher-order structure in perceptual representations. As an illustrative analysis, Fig. S15 plots pairwise correlations for different features in the generated samples (Exp. 2a). For example, we see that duration and F0 perturbation were significantly correlated for sadness ($r = .28$) but not for the other emotions (anger: $r = -.03$, happiness: $r = .00$); in contrast we see that pitch level and pitch slope were positively correlated for all three emotions. These kinds of higher-order analyses provide a more expressive perspective on prosody features than previous research, which mainly focuses on the independent contributions of single features rather than interactions between features. We intend to explore these kinds of interactions more in future research.

## Appendix E Musical chords

### E.1 Supplementary methods

This study applied GSP to the perceived pleasantness of musical chords. Each of these chords comprises three tones, and is hence termed a *triad*. We represented each triad as a pair of numbers, following the 'pitch chord type' representation of [17], which represents each chord tone as a pitch interval in semitones from the bass (i.e., lowest) tone. This representation captures the sense in which human pitch perception is relative (i.e., pitches are heard relative to their recent auditory context) and logarithmic (i.e., perceived pitch distance is approximately proportional to the difference in the logarithm of the frequencies) [18]. Integer values in this representation correspond to the standard 12-tone equal-tempered tuning system of Western music.

Figure S14: **A**: Average parameter settings for across- and within-participant chains in iteration 20 (Exp. 2a, 2b, 95% confidence intervals over chains). **B**: Mean validation contrast for different iterations (Exp. 2d, 95% confidence intervals over participants). Contrast is defined as the difference between the rating for the target emotion and the mean rating for the non-target emotions. **C**: Average parameter settings for all iterations in within-participant chains (Exp. 2a).

Figure S15: Pearson correlations between parameters in all three emotions (Exp. 2a).

We generated chords using Tone.js, a Javascript library for synthesizing sounds in the client's browser.[9] Each triad was synthesized as three simultaneous complex tones comprising 10 harmonics

Figure S16: Kernel density estimates generating the four sets of KDE modes considered in the validation experiment for musical triads (Exp. 3b). The top five modes are indicated in red. The density values are computed relative to a uniform distribution.

with amplitudes scaled by 12 dB/octave. These complex tones were presented with an ADSR envelope comprising a linear attack portion of 200 ms and a maximum amplitude of 1.0, an exponential decay portion lasting 100 ms taking the amplitude to 0.8, and a final exponential decay release portion lasting 1 s. The pitch of the bass tone was sampled uniformly and continuously in the logarithmic range G3–F4 (i.e., 196–349 Hz). The other two tones were specified by two continuous intervals in the range [0.5, 11], with the limits chosen such that the unison (0) and octave (12) were excluded, to prevent duplicating the pitch class of the bass tone.[10] We did however allow the two non-bass tones to overlap.

In each trial of the main experiment (Exp. 3a), participants were presented with the following prompt: 'Adjust the slider to match the following word as well as possible: pleasant'. Releasing the slider

Figure S17: An example trajectory of a GSP chain over chords, layered on top of a KDE of aggregated data from iterations 10–39 (bandwidth = 0.175 semitones, Exp. 3a). Similar dynamics were apparent in all other chains.

prompted a chord to be played whose pitch intervals reflected the current position of the slider. Before beginning the main experiment, participants completed three training examples to familiarize themselves with the procedure.

Through this experiment we constructed 50 across-participant chains comprising 40 iterations. The starting seed for each chain was created by randomly sampling both intervals from a uniform distribution in the range [0.5, 11]. Each participant contributed up to 20 trials to these chains.

The validation experiment (Exp. 3b) collected ratings for various raw samples and KDE modes, extracted from various iterations of the 50 chains constructed in the GSP experiment. The raw samples were represented by 16 experimental conditions corresponding to the iterations 0, 1, ..., 9 and 14, 19, ..., 39, with each condition being represented by 50 stimuli (1 stimulus from each chain). In addition to the raw sample conditions, there were four KDE conditions, with each KDE condition being represented by 5 modes, and the modes being extracted by aggregating data from iterations 0, 10 to 19, 10 to 29 and 10 to 39 (Fig. S16); here iteration 0 corresponds to the random seed. We used Gaussian kernels of width 0.175, and extracted modes using the kernel-based clustering algorithm of [19] as implemented in the ADPclust R package [20] with the number of clusters set *a priori* to 20, and keeping the five resulting centroids with the highest kernel density. In total, this resulted in 20 experimental conditions comprising 820 stimuli in total.

In each trial of the validation experiment, the participant was assigned to a randomly chosen stimulus from one of the conditions, and was asked to rate how pleasant that stimulus was on a four-level scale: 'Not at all', 'A little', 'Quite a lot' and 'Very much'. Overall we collected 662 ratings for each experimental condition, with each participant contributing up to 80 ratings.

We should note that while in Gibbs sampling it is customary to consider samples in jumps of full coordinate sweeps, here we decided to aggregate data continuously, given the inherent symmetry between the two intervals, so as to improve the quality of the estimated modes. We further exploited that symmetry by folding the data along the $x = y$ line, since reordering a pair of intervals does not alter the generated chord. Fig. S17 shows the raw data, where the $x = y$ symmetry is clearly apparent, and Fig. S16 shows the folded distribution after re-ordering the two intervals.

Figure S18: Combined marginal distributions for the two intervals, using a sliding window of length 20 (Exp. 3a).

## E.2 Supplementary results

In the main paper, we mostly discussed the structure and validation of features and raw samples aggregated across various chains and iterations. Fig. S17 complements this perspective by presenting the trajectory of a typical chain (Exp. 3a). It is clear that the dynamics are far from an optimization regime, where one would expect to see small and converging updates toward some local optimum (e.g., [21]). Instead, the trajectories illustrate the sampling regime of GSP, characterized by big leaps and lack of convergence, scanning the various regions of the space.

Fig. S18 shows the behavior of the combined marginal distributions for the two intervals, computed over a sliding window of length 20 with Gaussian kernels (bandwidth = 0.175 semitones, Exp. 3a). We see that two strong modes emerge at the perfect fifth (7) and the major third (4), alongside other peaks at integers and dips at the semitone (1) and tritone (6), reflecting the standard Western tonal hierarchy [22].

Audio samples of the top 15 KDE modes extracted from iterations 0 (random) and 10–39 can be found at `https://doi.org/10.17605/OSF.IO/RZK4S` in the folder `sound-examples-musical-triads`, with the modes arranged in descending order of density (`random_seed_top_modes_iter_0_0.wav` and `top_modes_iter_10_39.wav`).

## Appendix F   Faces

### F.1   Supplementary methods

This study used the 'StyleGAN' model of [23, 24] pretrained on the FFHQ dataset of faces from Flickr [23]. This model is a generative adversarial network, comprising a latent vector $\mathbf{z}$ sampled from a probability distribution $p(\mathbf{z})$, an input layer that takes a constant input $\mathbf{y}_0$, and the other layers $\mathbf{y}_i$ taking the previous layer and a non-linear function of $\mathbf{z}$ as an input:

$$\mathbf{y}_i = G_i(\mathbf{y}_{i-1}, \mathbf{w}), \quad \mathbf{w} = M(\mathbf{z}), \tag{6}$$

where $M$ is an 8-layer multilayer perceptron and the output layer $\mathbf{y}_L$ corresponds to an RGB image.

The study depended on participants interactively manipulating principal components of the **w** vector using a slider. We achieved this by creating an API that took as input a random seed for the latent vector **z**, a vector of principal component values for **w**, and the index of the principal component to be manipulated by the slider. The API then returned a video where the active principal component was incrementally modified through a specified number of standard deviations about the mean, with this API building on code released in [25]. The resulting video was then streamed to the participant's local computer, with the slider selecting between different frames of the video. We hosted the API on an AWS EC2 instance fitted with an NVIDIA K80 GPU.

An important technical issue concerned ensuring that participants didn't have to wait for the relatively slow stimulus generation process. We therefore generated stimuli asynchronously in advance of a given experimental trial, with participants being randomly assigned to the pool of currently available stimuli for each trial. Aggregating multiple responses per step of the GSP process helped in this regard, meaning that a higher throughput of participants could be sustained for a given rate of stimulus production.

The main experiment (Exp. 4a) evaluated six adjectives which we thought could elicit meaningful perceptual associations: 'attractive', 'fun', 'intelligent', 'serious', 'trustworthy', and 'youthful', with these choices informed by prior literature (e.g., [26]). Three across-participant chains were constructed for each of these adjectives, each of length 50 plus the initial random state, resulting in a total of 18 chains. Each step in the chain received five responses from five different participants, which were then aggregated using the arithmetic mean.

Participants were recruited from AMT as before with the stipulation that they be resident in the US. All participants were pre-screened with the color vocabulary task used previously for the color experiment. After completing a short demographic questionnaire, they took six practice trials to familiarize themselves with the task, then proceeded to the main experiment, where they completed up to 18 trials (one from each chain).

The validation experiment (Exp. 4b) recruited participants in the same manner, and had the participants rate all generated samples from iterations 1–10, 20, 30, 40, and 50. Each participant contributed 80 ratings, under the constraint that they never rated the same sample twice, and with participants being assigned to stimuli such that the number of ratings accumulated equally across stimuli. Data collection was continued until all samples had been rated at least 50 times.

We additionally conducted several follow-up GSP experiments to explore the paradigm further, described below and in Table S1:

1. We tested an alternative aggregation approach, where we summarized the five responses for each item with a KDE (Gaussian kernel, standard deviation of 0.5 in units of PCA standard deviations), and took the mode of the resulting distribution (Exp. 4c, Fig. S20).

2. We tested a small number of alternative methods for constructing a basis for the stimulus space (Exp. 4e, Fig. S24). In addition to the original PCA, we tested sparse PCA using a sparsity parameter of 1.0 (see the alpha parameter of `SparsePCA` from the `scikit-learn` package) and independent component analysis (ICA). We also tested the effect of retaining dimensions 71–80 instead of dimensions 1–10 of the PCA solution. In this experiment we only used the adjective 'attractive', and reduced the chain length to 30 iterations. For comparability with the original results of Exp. 4a, all chains were initialized to the same random seeds as in the original experiment.

3. We reran the original experiment but with the StyleGAN model pretrained on a dataset of faces from WikiArt (`https://www.wikiart.org`; `https://github.com/ak9250/stylegan-art`), to illustrate the dataset-dependence of the results (Exp. 4h, Fig. S29).

4. We reran the original experiment using KDE modes and relaxing participant recruitment to accept both US and non-US participants (Exp. 4i, Fig. S20). The resulting participant group was dominated by Indian (c. 50%) participants but also included a high proportion of US participants (c. 40%).

5. We reran the original experiment but asking participants to adjust the slider to 'find the person that you would most like to date', assigning self-reported male and female participants to separate chains so that we could perform a group-difference analysis (Exp. 4j, Fig. S23).

We additionally ran several rating experiments to complement these GSP experiments (Table S2). Exp. 4d collected ratings for the KDE mode experiment (Exp. 4c) and the global participant group experiment (Exp. 4i), as well as collecting ratings for the original experiment (Exp. 4a), with otherwise the same design as the original validation experiment (Exp. 4b), including the US-only criterion (Fig. S20). Exp. 4f used the same approach to collect ratings for the basis experiment (Exp. 4e, Fig. S24).

Exp. 4g used a similar design to investigate biases at different stages of the modeling pipeline (Fig. S25, S26, S27, S28). Stimuli were sourced from three stages: (a) random samples from the StyleGAN's FFHQ training dataset ($N = 300$); (b) random samples from the StyleGAN model ($N = 300$); random samples from the StyleGAN model, but only allowing the top 10 principal components to vary ($N = 300$); (c) samples from iterations 0, 10, 20, 30, 40, and 50 of the GSP processes from Exp. 4a ($N = 108$). Instead of asking participants to rate how well the images matched the GSP adjectives, we instead asked participants to answer questions from the following list:

1. What is the gender of the person in the image?
2. Is the person in the image of white ethnicity?
3. Is the person in the image smiling?
4. Is the person in the image wearing a hat?
5. Is the person in the image wearing formal clothes?
6. Is the person in the image wearing glasses?

In each case, the participant was presented with three options: "Male"/"Female"/"Other" in the case of gender, and "Yes"/"No"/"Don't know" in the other cases. We also asked participants to estimate the age in years of the person depicted in the image.

We had two kinds of motivations for choosing these particular evaluations. We chose gender, age, and ethnicity because these are two criteria according to which many people experience bias in the real world, and we wanted to understand how these variables were treated by the modeling pipeline. We chose the other four evaluations because they are examples of easily quantified features that seem likely to influence judgments made about the person. Of course, it should be acknowledged that some of these variables are impossible to determine definitively from an image; for example, it is a substantial simplification to treat gender and ethnicity in a categorical way. However, we anticipated that this simplification would be necessary to make the task understandable to the participants, and that the resulting data would nonetheless be informative about the kinds of biases present in the modeling pipeline.

## F.2   Supplementary results

**Raw samples from the main experiment.** Example raw samples and validation results from the main experiments (Exp. 4a, 4b) are shown in Fig. 4 of the main paper. Fig. S19 illustrates the raw samples in more detail, displaying iterations 0–10, 20, 30, 40, 50 from one chain for each target word. It is clear from both the validation results and the raw samples that the chains make clear progress towards the target category already by the end of the first sweep (10 iterations), and sometimes the resemblance to the target category does not improve noticeably after this point. However, this does not mean that the process converges to a static image after this point: instead, there is a moderate amount of variety in the subsequent faces (see also Fig. S21 and S22). The process is therefore still somewhat in the stochastic sampling regime rather than the deterministic optimization regime.

**Validation results for follow-up experiments.** Fig. S20 plots validation results for Exp. 4a (mean aggregation, US-only participants), Exp. 4c (aggregation with KDE modes, US-only participants), and Exp. 4i (aggregation with KDE modes, global participants), as collected in Exp. 4d. The broad trends in the ratings are replicated across the three experiments: typically almost all of the improvement comes in iterations 1–10, with ratings staying mostly stable after this point. All three experiments struggle to capture trustworthiness, which is clearly a particularly subjective judgment to make. Interestingly, there is no evidence that KDE peak-picking outperforms the arithmetic mean as an aggregation technique. Inspecting the raw data and the density estimates, this does not seem to be a consequence of poorly chosen kernel width or artifacts in the density estimation process. Instead, it seems that the participants' conditional distributions could typically be approximated well by a unimodal distribution, and hence averaging was a sensible aggregation method.

## Iteration

Figure S19: Raw samples from six GSP chains in Exp. 4a (US-only participants, mean aggregation).

Figure S20: Validation results for Exp. 4a, 4c, and 4i, as produced in Exp. 4d. The shaded regions correspond to 95% confidence intervals over participants.

**Hints at cross-cultural differences.** Raw samples of six chains from Exp. 4c (aggregation with KDE modes, US-only participants), and Exp. 4i (aggregation with KDE modes, global participants) are displayed in Fig. S21 and S22. It is important not to read too much into these raw samples, as they ultimately come from stochastic distributions and will vary over repeated runs. However, we did notice some suggestive differences between the final samples of the US chains and those of the global chains. Most salient was the fact that all US chains for 'intelligent' finished with a Caucasian man, whereas the three final states of the global chains included both a woman and a non-Caucasian man. We also noticed that the global chains were the only ones to include a man as the final 'attractive' sample. While some of this variation will be due to chance, the remaining variation will presumably reflect different stereotypes held by the different participant groups. It would be interesting to explore these different stereotypes in more systematic ways.

Figure S21: Raw samples from six GSP chains in Exp. 4c (US-only participants, KDE mode aggregation).

**Gender differences.** Exp. 4j provides a second proof of concept for this kind of group-difference approach (Fig. S23). Here participants were split by self-reported gender, and instructed to optimize the slider for a person that they would most like to date. As one might expect, the samples reflect a predominant (but not universal) preference for members of the opposite gender. This in itself may be a trivial result, but it is easy to intuit how one could extrapolate this approach to much more complex and interesting group-difference studies, for example those involving different cross-cultural populations.

**Basis construction methods.** Fig. S24 plots validation results for Exp. 4e (exploring different basis construction methods), as collected in Exp. 4f. The results suggest an early advantage for the original PCA technique; however, the discrepancy with sparse PCA and ICA is small, and seems to disappear after more iterations. As would be expected, the version of PCA with components 71–80 performs poorly; in practice, these components contribute very little perceptually speaking (see also [25]). On this basis, there is little evidence to dismiss any one of PCA, sparse PCA, or ICA. Future work should also consider other recently proposed approaches for parameterizing the generative model, for example [27, 28].

**Bias analyses.** Fig. S25 plots perceived gender in the different datasets evaluated in Exp. 4g. We see that the gender balance is fairly equal between men and women, with perhaps slightly more women than men as the pipeline progresses. Fig. S26 plots perceived age as a function of perceived gender in the same datasets. Looking first at the training dataset, we see that the mean age is close to 30 years, with the male faces tending to be perceived as somewhat older than 30, and the female faces being perceived as slightly younger than 30. This association between age and gender is amplified

**Iteration**

Figure S22: Raw samples from six GSP chains in Exp. 4i (global participant group, KDE mode aggregation).

Figure S23: Final samples from the first four male and female chains in the dating preferences experiment (Exp. 4j).

to a certain amount through the modeling pipeline, even before the PCA process; it seems as if the model is capturing this association and stereotyping it to a certain degree. This relationship has interesting implications for the GSP samples; if female samples tend to be subjectively younger than male samples, and if younger faces tend to be perceived as more attractive, then GSP samples for 'attractive' will be biased towards women, even if the participants do not possess any systematic bias for women over men. Likewise, if older faces tend to be perceived as more intelligent, then this relationship between age and gender would be expected to induce a bias in the GSP samples for 'intelligent' towards male faces.

There are many other similar biases that one could anticipate affecting the GSP process. To illustrate some of these potential biases, Fig. S27 plots judgments for ethnicity, smiling, hats, formal clothes, and glasses wearing for the four datasets in Exp. 4g, split by gender. We see for example that men are much more likely than women to be portrayed in formal clothes, potentially a further reason why 'intelligent' GSP samples tend to favor men. Similarly, men are more likely to be portrayed in glasses, another potential contributor to perceived intelligence. Conversely, women are more likely than men to be smiling, potentially supporting a female bias in the 'attractive' and 'fun' GSP samples. These hypotheses are consistent with Fig. S28, which shows that perceived intelligence is indeed associated

Figure S24: Validation results for Exp. 4e (exploring different basis construction methods), as collected in Exp. 4f. The shaded regions correspond to 95% confidence intervals over participants.

Figure S25: Perceived gender for faces from different stages of the modeling pipeline, as collected in Exp. 4g.

Figure S26: Perceived age split by gender for faces from different stages of the modeling pipeline, as collected in Exp. 4g. The error bars denote 95% confidence intervals bootstrapped over images.

Figure S27: Evaluations of ethnicity, smiling, hats, formal clothes, and glasses, for faces from different stages of the modeling pipeline, split by gender (Exp. 4g). The error bars denote 95% confidence intervals bootstrapped over images.

with wearing formal clothes and glasses, and that perceived attractiveness and fun are both associated with smiling. These examples illustrate the complex network of biases that can be inherited from a generative model such as StyleGAN, and highlight the importance of developing more balanced training datasets for future cognitive work in this area.

**Training dataset.** To provide a more intuitive illustration of the method's dependence on the training dataset, Fig. S29 displays final GSP samples from Exp. 4h, which used the StyleGAN model trained on a dataset of portraits from WikiArt (`https://www.wikiart.org/`). The artistic nature of the WikiArt dataset differs clearly from the photographic nature of the FFHQ dataset, and this is reflected in the GSP samples. Nonetheless, the GSP process still successfully navigates this new space to find samples that subjectively reflect the target adjectives.

Figure S28: Evaluations of ethnicity, smiling, hats, formal clothes, and glasses, for GSP samples evaluated in Exp. 4g. The error bars denote 95% confidence intervals bootstrapped over images.

Figure S29: Final GSP samples from Exp. 4h, which used the StyleGAN model pretrained on a dataset of portraits from WikiArt (https://www.wikiart.org/).

### F.3  Conclusion

Our analyses indicate that GSP is an effective tool for exploring the generative space of the StyleGAN model. Here we relied on a simple PCA approach for creating a reduced basis of the generative space, but there are other promising approaches in the literature that could also be applied to this task (e.g., [27, 28]). However, our analyses also indicate that dataset bias is a real and important issue when interpreting the outcomes of this approach. Future work must engage with this problem by studying the kinds of biases inherent in their generative models and ideally finding ways to construct less biased models in the first place (e.g., [29]).

## Appendix references

[1]  Y. Weiss, E. P. Simoncelli, and E. H. Adelson, "Motion illusions as optimal percepts," *Nature Neuroscience*, vol. 5, no. 6, pp. 598–604, 2002.

[2]  X.-X. Wei and A. A. Stocker, "A Bayesian observer model constrained by efficient coding can explain 'anti-Bayesian' percepts," *Nature Neuroscience*, vol. 18, pp. 1509–1517, 2015.

[3]  A. N. Sanborn, T. L. Griffiths, and D. J. Navarro, "A more rational model of categorization," in *Proceedings of the 28th Annual Conference of the Cognitive Science Society* (R. Sun and

N. Miyake, eds.), pp. 726–731, Cognitive Science Society, 2006.

[4] D. McFadden, "Conditional logit analysis of qualitative choice behaviour," in *Frontiers in Econometrics* (P. Zarembka, ed.), pp. 105–142, New York, NY: Academic Press, 1974.

[5] K. E. Train, *Discrete choice methods with simulation*. Cambridge University Press, 2009.

[6] K. J. Woods, M. H. Siegel, J. Traer, and J. H. McDermott, "Headphone screening to facilitate web-based auditory experiments," *Attention, Perception, & Psychophysics*, vol. 79, no. 7, pp. 2064–2072, 2017.

[7] M. Chmielewski and S. C. Kucker, "An MTurk crisis? Shifts in data quality and the impact on study results," *Social Psychological and Personality Science*, vol. 11, no. 4, pp. 464–473, 2020.

[8] J. H. Clark, "The Ishihara Test for color blindness," *American Journal of Physiological Optics*, vol. 5, pp. 269–276, 1924.

[9] "IEEE recommended practice for speech quality measurements," tech. rep., IEEE, 1969. ISBN: 9781504402743.

[10] P. Demonte, "HARVARD corpus speech shaped noise and speech-modulated noise for SIN test," 2019. Publisher: University of Salford.

[11] P. Boersma and D. Weenink, "Praat: doing phonetics by computer [Computer program]." Version 6.0.37, `http://www.praat.org/`, 2018.

[12] Y. Jadoul, B. Thompson, and B. de Boer, "Introducing Parselmouth: A Python interface to Praat," *Journal of Phonetics*, vol. 71, pp. 1–15, 2018.

[13] I. R. Titze, Y. Horii, and R. C. Scherer, "Some technical considerations in voice perturbation measurements," *Journal of Speech, Language, and Hearing Research*, vol. 30, no. 2, pp. 252–260, 1987.

[14] R. Corretge, "Praat vocal toolkit." http://www.praatvocaltoolkit.com, 2020.

[15] A. S. Cowen, P. Laukka, H. A. Elfenbein, R. Liu, and D. Keltner, "The primacy of categories in the recognition of 12 emotions in speech prosody across two cultures," *Nature Human Behaviour*, vol. 3, no. 4, pp. 369–382, 2019.

[16] P. Laukka, H. A. Elfenbein, N. S. Thingujam, T. Rockstuhl, F. K. Iraki, W. Chui, and J. Althoff, "The expression and recognition of emotions in the voice across five nations: A lens model analysis based on acoustic features," *Journal of Personality and Social Psychology*, vol. 111, no. 5, 2016.

[17] P. M. C. Harrison and M. T. Pearce, "Representing harmony in computational music cognition," *PsyArXiv*, 2020.

[18] T. Stainsby and I. Cross, "The perception of pitch," in *The Oxford handbook of music psychology* (S. Hallam, I. Cross, and M. Thaut, eds.), pp. 47–58, New York, NY: Oxford University Press, 2009.

[19] A. Rodriguez and A. Laio, "Clustering by fast search and find of density peaks," *Science*, vol. 344, no. 6191, pp. 1492–1496, 2014.

[20] X.-F. Wang and Y. Xu, "Fast clustering using adaptive density peak detection," *Statistical methods in medical research*, vol. 26, no. 6, pp. 2800–2811, 2017.

[21] N. Jacoby and J. H. McDermott, "Integer ratio priors on musical rhythm revealed cross-culturally by iterated reproduction," *Current Biology*, vol. 27, no. 3, pp. 359–370, 2017.

[22] C. L. Krumhansl and E. J. Kessler, "Tracing the dynamic changes in perceived tonal organization in a spatial representation of musical keys," *Psychological Review*, vol. 89, no. 4, pp. 334–368, 1982.

[23] T. Karras, S. Laine, and T. Aila, "A style-based generator architecture for generative adversarial networks," in *Proceedings of the IEEE Conference on Computer Vision and Pattern Recognition CVPR*, pp. 4401–4410, 2019.

[24] T. Karras, S. Laine, M. Aittala, J. Hellsten, J. Lehtinen, and T. Aila, "Analyzing and improving the image quality of StyleGAN," *arXiv*, 2019.

[25] E. Härkönen, A. Hertzmann, J. Lehtinen, and S. Paris, "GANSpace: Discovering interpretable GAN controls," *arXiv*, 2020.

[26] L. Brinkman, A. Todorov, and R. Dotsch, "Visualising mental representations: A primer on noise-based reverse correlation in social psychology," *European Review of Social Psychology*, vol. 28, no. 1, pp. 333–361, 2017.

[27] A. Voynov and A. Babenko, "Unsupervised discovery of interpretable directions in the GAN latent space," *arXiv*, 2020.

[28] Y. Shen and B. Zhou, "Closed-form factorization of latent semantics in GANs," *arXiv*, 2020.

[29] A. Grover, K. Choi, R. Shu, and S. Ermon, "Fair generative modeling via weak supervision," *arXiv*, 2019.

## Footnotes

[3]`https://github.com/Dallinger/Dallinger`

[4]`https://www.heroku.com/home`

[5]`https://aws.amazon.com/`

[6]https://www.mturk.com

[7]Age and gender distributions were computed from all participants who passed the pre-screening tasks, excluding the validation experiments for emotional prosody, for which demographic information was not collected. Participant numbers only include participants who contributed at least one valid trial to the main experiment.

[8]An experimental condition typically corresponded to one point on a figure, for example the third iteration for the second 'lavender' GSP chain.

[9] https://tonejs.github.io/

[10]Two tones are said to share the same pitch class if they are separated by an integer multiple of 12 semitones (an octave).