[Reviews · NeurIPS 2020]

Review 1

Summary and Contributions: The manuscript introduction Gibbs sampling with people, a method for exploring people’s perceptual spaces. The approach is an extension of Markov chain monte carlo with people such that binary decisions are relaxed to a choice to select from a continuous dimension. The new approach is compared with MCMCMP, showing distinct improvements. Results further demonstrate GSP on three additional domains, and in combination with image synthesis (StyleGAN) and interpretable networks (GANSpace) to illustrate exploration of high dimensional latent spaces.

Strengths: Exploring human perceptual representations is an important problem that arises in many contexts. Existing methods are limited by the guess-and-check nature of exploration eliciting people’s judgements. MCMCP was a promising approach because it relieved the problem of having to systematically explore the space by creating a markov chain of decisions. However, simple binary judgements and the continued need to guess and check (via the proposal distribution) limited applicability. The generalizations presented here make the approach considerably more general, as demonstrated in the experiments. The empirical evaluation is impressive in scale, scope and results.

Weaknesses: Of the theoretical questions, it was particularly nice to see the probability matching interpretation of human decisions was engaged directly. One potential challenge for the utility theoretic formulation is the well-documented phenomenon in which goodness or typicality (more generally, utility judgments) vary across categories. On the one hand, I suppose one could argue the approach is to estimate that, on the other, this no longer necessarily reflects the underlying perceptual space. For example, taller trees are deemed more typical, even though they are statistical outliers. It is hard to say for sure to what degree this is a feature (reflecting quirks of human judgments) or a bug (doing so in a way that misses important statistical and representational aspects of experience). It would be nice to see aggregation applied to MCMCP in the main text as well. The results do seem to suggest that GSP is a bit better, but the real power seems to come from aggregating. As noted in the text parameterizing the space seems important to the effectiveness of the method. It would be nice to have demonstrations of the consequences of different parameterizations and some thoughts on how to this might be done well in general (is PCA really effective?).

Correctness: The analysis seems correct and complete.

Clarity: The paper is very well written.

Relation to Prior Work: The paper reviews prior work (primarily MCMCP, but also others) and clearly demonstrates that the proposed method is an improvement.

Reproducibility: Yes

Additional Feedback: ===Post response==== I have no further updates to my review. I don't know if this will be surfaced to the authors, but the paper was flagged for as a potential ethics concern. I do not concur with that opinion, but it would be helpful if the authors could try to sharpen that point that the method is aimed at understanding biases rather than creating datasets for training machine learning algorithms.


Review 2

Summary and Contributions: - The authors propose Gibbs sampling with people, an alternative to the ‘mcmc with people’ idea for sampling participants ‘mental representations. - A comparison is made between MCMCP and GSP though several experiments, with follow-up experiments examining specific aspects of the method such as aggregation

Strengths: - This is a very nice idea, backed up by a series of experiments that each expand on the ideas in the paper - The writing is very good and the figures clear - The background is well described, and the theory behind the method is described in a good amount of detail - This is certainly very relevant to the NeurIPS community and very novel

Weaknesses: - Some details of the experiments are missing, but that is to be expected given the number of experiments. The appendix clarified my questions

Correctness: - This look all perfectly correct

Clarity: - Clear and well written

Relation to Prior Work: - Nicely builds on previous work (MCMCP)

Reproducibility: Yes

Additional Feedback: - One question that still remained after reading the main text (although partly answered by the appendix) was regarding the total number of responses for GSP and aggregated GSP. If subjects have to make 10 responses per iteration, any benefit per response may be gone? - Related: how do the experimenters avoid subjects merely making the same response 10 times? Just by varying the starting position of the slider? - Devil’s advocate: Might be good to talk about how this differs from e.g. a multidimensional methods of adjustments in psychophysics?


Review 3

Summary and Contributions: This paper describes a new method, Gibb's Sampling with People (GSP), for exploring people's semantic representations. The technique builds on MCMC with people but uses a continuous choice, rather than a binary choice, for selecting the next stimulus in the chain. Theoretically, there are two main contributions: (a) reframing MCMCP in way that doesn't require that people be probability matching (rather than maximizing), and (b) developing the GSP approach. Within both of these, participants are assumed to extract a utility value, made up of an actual utility and a noise component, and the behavior of the sampler is contingent on the scale of the noise component. The paper considers several methods of aggregating samples in order to reduce noice and make it easier to estimate the mode of a distribution. To test GSP, the paper presents a number of studies (4 in the main paper, plus additional variations in the supplemental materials), and establishes that GSP is more effective at mode-seeking than MCMCP, especially with aggregation, and can uncover interesting information about people's semantic representations of categories and continuous perceptual spaces.

Strengths: One of the major strengths of this pape is the breadth of the empirical results, which both serve to illustrate the strengths of the new methodology and apply that methodology in interesting ways. By looking at four different domains, ranging from the relatively simple color stimuli to the high dimensional face stimuli, the experiments are able to go beyond simply showing that GSP has advantages to MCMCP to showing what sort of insights might be gained by exploring complex representations in this way and offering some suggestive evidence for being able to uncover cross-cultural differences in representations and stereotypes. A second strength is in the reframing of MCMCP to not require the assumption that people probability match. Finally, I believe GSP is likely to be adopted by psychology and cognitive science researchers interested in representations, leading to a potentially large impact of this work. While the material is in some ways different from the "typical" NeurIPS paper, I think there is likely to be substantial interest by those who are interested in representations generally as well as those interested in cognitive science specifically.

Weaknesses: Overall, I thought this was a strong paper. The main concerns I had were as follows: (1) Mode-seeking versus showing the distribution: The aggregated results in the first experiment seem to show much more homogeneity than the results for GSP or MCMCP. It seems like one limitation of this approach might be that there is limited exploration of the space, perhaps making it hard to move between modes, and also makes it more difficult to see the full shape of the distribution, which I have often taken to be a goal in work using MCMCP. The movement between optimization and seeking a distribution is discussed to some extent in the paper, but I would be interested in seeing this discussed more (and perhaps whether GP without aggregation is likely to lead to more optimization than MCMCP). In the author response, they have shown additional information suggesting that GSP is more mode-seeking but also does a better job of capturing the distribution. While this doesn't completely get at cases that are more multimodal than colors, it does go a long way to addressing this concern and I look forward to reading more details about the new experiment in the supplement. (2) Possible conflation of participant representations and representations in the stimuli space: In the final experiment, the faces are generated via a GAN and manipulated by automatically generated axes in the space of faces generated by the GAN. The stimuli space thus seems far more subjective and dataset-dependent than in the other cases, and to draw conclusions about people's representations of faces from this, I would want to see additional chains from a GAN trained with a different dataset of images (and perhaps to get some sense of how similar the dimensions are across different training sets). That isn't the main point of this paper, and thus I don't see it as a prohibitive weakness, but it does make me concerned about interpreting the results of that final experiment and whether it would be possible to really gain insights into people's representations about the relevant categories. (And these problems seem likely to be made worse by being in an optimization regime, as that experiment is due to the aggregation - the conclusion that there isn't much multimodality around these concepts also seems surprising and understanding how the results across chains were compared would be helpful). The author response reframes this experiment a bit and also adds an experiment with an additional dataset, which is helpful. Their response makes clear that they are planning to reframe the discussion of the experiment in the paper a bit as well, and I would encourage them to be clear about limitations in terms of assumptions of a single representation for particular concepts: it seems unlikely that people monolithically have a single distribution over concepts like "attractive" and to the extent that Turkers tend to under-represent Black and Latinx populations in the US (at least based on 2016 Pew Research results - I'm not sure about follow up work), it seems like the favoring of Caucasian faces for particular concepts should be placed in a broader context.

Correctness: The claims and method in the paper appear to be correct.

Clarity: Overall, the paper was clearly written. Some of the information about aggregation was a little hard to understand in an initial read through, but the supplementary materials provide additional information that made these parts more clear.

Relation to Prior Work: The paper made clear how this was related to prior work and described what contributions were new.

Reproducibility: Yes

Additional Feedback: My main feedback is summarized above. I thought the work was very interesting, and while the difference in setup is relatively simple, this is a good methodological innovation. As noted above, the face experiment is the one that I find the most dubious, and that's both in terms of the reliance on the underlying dataset and using the results to draw conclusions about people's representations. While I know space is limited, some discussion about what conclusions can be drawn from the results (beyond "here's an example of a trustworthy/intelligent/etc face") would I think be really helpful for illustrating the benefits of the proposed approach for studying high dimensional stimuli to researchers in cognitive science (of which I consider myself one). As expanded upon a bit above, this is also a place where I think there is potential for misunderstanding that perpetuates societal biases; I appreciate the attention to these issues in the Broader Impact statement, and I think bringing some of these ideas into the main paper when discussing this experiment and/or limitations more generally would be very helpful for making clear that these representations are not speaking to intrinsic qualities of the stimuli but about the representations of the people in the study (and that the focus is on the modes, which may mean that representations of subpopulations in the study are even less likely to be represented in the final distribution). While I recognize that any study can be misconstrued, I think making the specifics here very clear could diminish the possibility of this being a paper that is used to reinforce racial and gender stereotypes.

[Author Response · NeurIPS 2020]

We have addressed the reviewers' comments by running seven new experiments, which shed useful new light on some of these issues. We have updated the main text and appendices to respond to each comment, reporting the new experiments and adding further discussion where appropriate.

*R2: The authors propose that MCMCP/GSP estimate utility, whereas Sanborn & Griffiths proposed that MCMCP estimates subjective probability; however, it is not clear whether utility and subjective probability are equivalent, and which should be preferred. A:* To investigate whether these possibilities can be differentiated, we reran the MCMCP and GSP color tasks with three different questions probing different constructs. However, we found no clear effects of question type, suggesting that the three constructs can be equated here (Fig. A).

*R2: How about aggregated MCMCP?* We reran the color experiment with aggregated MCMCP and found that aggregated GSP performed worse than both forms of GSP (Fig. B).

*R2: GSP seems intuitively dependent on parametrization, can you discuss?* To address this issue we reran the face experiment comparing three parametrization methods (PCA, sparse PCA, ICA), as well as low-variance components of PCA as a control. We found good performance in each case except the low-variance components, suggesting that several data-driven methods can recover sufficiently psychologically meaningful dimensions for GSP (Fig. C).

*R3: Does the benefit of aggregation disappear once you take into account the number of responses required? A:* We find that non-aggregated GSP performs best for short chains, but aggregated GSP performs best for long chains (Fig. S12).

*R3: How do the experimenters avoid subjects merely making the same response 10 times? A:* The across-participants algorithm ensures that no subject sees the same question twice; multiple chains are run in parallel (Fig. S5), and each participant only visits the same chain once.

*R3: It would be worth discussing how the technique differs from e.g. multidimensional methods of adjustment in psychophysics.* A: Our revised paper explains how our adaptive procedure differentiates GSP from slider paradigms common in psychophysics, which are tailored to identifying perceptual limits in low-dimensional spaces.

*R5: The theory section discusses the trade-off between mode seeking and stochastic sampling, but this trade-off is neglected when discussing Study 1. A:* We conducted a new experiment to derive a ground truth for the utility function, and designed a new analysis to examine the trade-off between mode-seeking and stochastic sampling. We confirm that GSP is more mode-seeking than MCMCP, but nonetheless recovers the utility function more reliably (Fig. D).

*R5: The authors changed the MCMCP trial question slightly from Sanborn & Griffiths (2008), I'm curious to see whether this had any effect on behavior.* We reran MCMCP and GSP on the color task, using three different questions including one closely resembling the Sanborn & Griffiths question. There was no systematic difference in outcomes here, supporting the notion that all these questions probe a common utility function (Fig. A).

*R5: The authors do not sufficiently acknowledge how biases in the GAN's training data affect the samples generated by the GSP process. To draw conclusions from the results, I would want to see additional results using a different training dataset.* Our revised manuscript acknowledges this by reframing GSP as a tool for navigating and interpreting the parameter space of generative models using participant judgements. To illustrate this, we conduct two further experiments, one manipulating the dataset (portraits vs. photographs, Fig. E) and one manipulating the participant group (male vs. female, Fig. F).



[Meta-Review · NeurIPS 2020]

This paper introduces a new method for eliciting human representations of perceptual concepts, such as what RGB values people think correspond to the color “sunset” or what auditory dimensions (e.g. pitch, duration, intensity) they think make a spoken sentence sound “happy” vs. “sad”. Rather than eliciting representations via guess-and-check (i.e., start with a dataset and then apply human-generated labels), this method (Gibbs Sampling with People, or GSP) enables inference to go in the other direction (i.e., start with labels, and then identify percepts that match those labels). GSP extends prior work (MCMC with People) to allow eliciting representations of much higher-dimensional stimuli. The reviewers unanimously praised this paper for tackling an important and relevant problem in cognitive science, for its breadth of empirical results, and for its novelty over prior work. R2 stated that the paper is “impressive in scale, scope, and results”, R3 stated that it was “very relevant to the NeurIPS community and very novel”, and R4 felt there could be “a potentially large impact of this work” with “substantial interest” amongst the NeurIPS community. I agree this work is very exciting, novel, and impactful, both for cognitive scientists and to machine learning researchers interested in representation. However, the paper does raise some cause for worry from an ethics standpoint. In eliciting percepts that match participants’ internal representations, GSP will naturally reproduce biases and stereotypes that those participants hold. Indeed, this can be seen in the results in one of the experiments in the paper, in which the method is used to elicit facial representations for attributes like “attractive”, “fun”, “intelligent”, etc., where the faces elicited are overwhelmingly white and reflect gender stereotypes such that men are “intelligent”. As such, this paper was flagged for ethics review and received three additional reviews, which I summarize here. The opinions of the ethics reviewers ranged substantially. ER3 felt that there was “potential concern regarding the perpetuation of racial and gender stereotypes” but that this was sufficiently addressed by the broader impact statement. ER1 noted that this work is a “very exciting approach for eliciting human semantic representations” but felt that the broader impact statement was insufficient and that “the risks coming from human and societal biases are understated, and I worry practitioners may build datasets or generative models that strongly encode these biases”. They requested three revisions (see below). ER2 was concerned that “drawing inferences on emotional states of a person based on perceptual judgments of listeners as well as the assumption that social and psychological characteristics...are something that can be read off of faces has a dark history in physiognomy”. They were also concerned that the paper “fail[ed] to mention how [social stereotypes being perpetuated] might be mitigated or how in fact such work can be of a net positive value to society and specifically to the ethnically and racially marginalized”. ER2 argued for rejection, and while I disagree with this recommendation, I do think that their assessment highlights the fact that the paper does not state clearly enough what the method is meant for (e.g. as a tool for psychologists), what it is not meant for (e.g. generating datasets for ML models) and why, and what the experimental limitations are. ER1's requested revisions (directly copied from their review): --- 1/ Make a much stronger and categorical assertion that the GSP (Gibbs sampling with people) method proposed will reflect individual and cultural biases. The Broader Impact Statement alludes to observed biases (towards men for 'intelligence' and towards women for 'beauty') but it doesn’t quantify them, nor explain what part of the bias is attributable to individuals vs. the composition of the dataset. There are very simple tests that can be done to test for both effects, and it seems at least that controlling for bias in the dataset should be a reasonable requirement. 2/ Call out the risk that this method will be used to generate datasets that reinforce and amplify existing stereotypes in society. What these stereotypes will look like will depend crucially on whom the “people” are in GSP. The paper uses Amazon Mechanical Turk to recruit people, but doesn’t delve into the limitations of this approach for recruiting diverse sets of “people.” It’s imperative to call out the stereotype and representation harms that can come from using non-diverse humans in GSP. My major concern is with stereotyping in the human faces example, but I’m also concerned about the music example. The fact that the distribution peaks at “prototypical sonorities from Western music” is another clear example of how biased the humans behind GSP can be, and the dangers of generating biased datasets. 3/ As a corollary of the two points above, call out the risks of using GSP for tuning the parameters or hyperparameters of generative deep nets. In the human faces example, there’s a direct connection to tuning the hyperparameters of a deep net for face generation, where if the input was set to “intelligent” you’d get more males. It’s imperative to demand that applications of GSP for tuning generative deep nets fully analyse the diversity of the humans behind GSP, and that they analyze their potential biases first. --- I am recommending conditional acceptance of the paper, as I would like the paper to be revised for the camera ready in light of the ethics reviews. In particular, I think the paper needs to do a better job at: (1) motivating the use case for this method and for the particular experiments from a cognitive science perspective (not just in the broader impacts but also throughout the paper); (2) making it clear that the elicited “representations are not speaking to intrinsic qualities of the stimuli but about the representations of the people in the study” (as suggested by R5); and (3) discussing ways in which GSP should not be used (such as generating datasets or fine-tuning ML models). I would also like the authors to (4) pay particular attention to (and address) R5’s suggestions in Q3 and Q8 and the suggestions from ER1 above. **Please note that acceptance of the paper is conditional on these four changes being made in the camera-ready.** ******************************* Note from Program Chairs: The camera-ready version of this paper has been reviewed with regard to the conditions listed above, and this paper is now fully accepted for publication.